# Rational engineering of an elevator-type metal transporter ZIP8 reveals a conditional selectivity filter critically involved in determining substrate specificity

Yuhan Jiang[1], Zhen Li[1], Dexin Sui[2], Gaurav Sharma[1], Tianqi Wang[2], Keith MacRenaris[3], Hideki Takahashi [2], Kenneth Merz[1,2] & Jian Hu [1,2✉]

Engineering of transporters to alter substrate specificity as desired holds great potential for applications, including metabolic engineering. However, the lack of knowledge on molecular mechanisms of substrate specificity hinders designing effective strategies for transporter engineering. Here, we applied an integrated approach to rationally alter the substrate preference of ZIP8, a Zrt-/Irt-like protein (ZIP) metal transporter with multiple natural substrates, and uncovered the determinants of substrate specificity. By systematically replacing the differentially conserved residues with the counterparts in the zinc transporter ZIP4, we created a zinc-preferring quadruple variant (Q180H/E343H/C310A/N357H), which exhibited largely reduced transport activities towards $Cd^{2+}$, $Fe^{2+}$, and $Mn^{2+}$ whereas increased activity toward $Zn^{2+}$. Combined mutagenesis, modeling, covariance analysis, and computational studies revealed a conditional selectivity filter which functions only when the transporter adopts the outward-facing conformation. The demonstrated approach for transporter engineering and the gained knowledge about substrate specificity will facilitate engineering and mechanistic studies of other transporters.

[1] Department of Chemistry, Michigan State University, East Lansing, MI 48824, USA. [2] Department of Biochemistry and Molecular Biology, Michigan State University, East Lansing, MI 48824, USA. [3] Department of Microbiology & Molecular Genetics, Michigan State University, East Lansing, MI 48824, USA. ✉email: hujian1@msu.edu

Transporter engineering has gained traction in recent years because of its great potential in broad applications, particularly in metabolic engineering[1–3]. As membrane transporters govern the fluxes of substrates, intermediates, or products across the cell and organelle membranes, transporter engineering has been viewed as a promising strategy to control the kinetics of reactions inside the biofactories. Metal transporters hold a unique position in dealing with beneficial and/or toxic metal uptake, distribution, accumulation, and extrusion. Engineering of metal transporters may reduce contamination of toxic metals in foods, enable biofortification, or facilitate phytoremediation[4]. Albeit these potentials, there were only limited trials on a few metal transporters to alter substrate specificity. These early studies were either only focused on a small set of residues or based on error-prone PCR facilitated random mutations[5–8].

One prominent target for transporter engineering is the Zrt-/Irt-like protein (ZIP) family (SLC39A), which selectively transports divalent *d*-block metals. Recent structural, biochemical, and computational studies have provided evidence supporting that the ZIP metal transporters utilize an elevator-type transport mode to achieve alternating access[9–11]. In the two-domain architecture of a representative ZIP from *Bordetella bronchiseptica* (BbZIP), the only ZIP whose transmembrane domain structure has been solved to date, transmembrane helices (TMs) 2/3/7/8 form the static and dimeric scaffold domain whereas TM1/4/5/6 form the transport domain which slides vertically as a rigid body against the scaffold domain during transport. As the transport sites (M1 and M2) are nearly exclusively located in the transport domain, they are alternately exposed to either side of the membrane when the transport domain shuttles across the membrane. Although several structures representing the inward-facing conformation (IFC) of BbZIP have been reported, an experimentally solved outward-facing conformation (OFC) of BbZIP is still missing. Nevertheless, an OFC model of BbZIP was generated by using the approach of repeat-swap homology modeling, which has been validated by biochemical approaches[11]. As an ancient protein family ubiquitously expressed in all kingdoms of life, the ZIPs are diverse in multiple aspects, including substrate specificity. Even with the gradually elucidated elevator transport mechanism, which is believed to be applicable to most if not all members of the ZIP family, how ZIPs selectively transport certain *d*-block metals while repelling the others remains elusive. Some ZIP family members appear to be specific toward zinc ions ($Zn^{2+}$), but the broader substrate spectrum of other family members allows them to function as multi-metal transporters capable of transporting a panel of divalent *d*-block metal ions. For instance, while most of the fourteen human ZIPs are reported to transport $Zn^{2+}$ and play roles in Zn homeostasis and Zn signaling[12–15], ZIP8 and its close homolog ZIP14 transport not only $Zn^{2+}$ but also ferrous ions ($Fe^{2+}$), manganese ions ($Mn^{2+}$), and cadmium ions ($Cd^{2+}$), and as such are critically involved in Fe and Mn homeostasis and are responsible for cellular Cd uptake and toxicity[16–28]. In *Arabidopsis thaliana*, Cd accumulation can be partially attributed to the $Cd^{2+}$ transport activity, in addition to the activities toward $Zn^{2+}$, $Fe^{2+}$, and $Mn^{2+}$, of the root-expressing IRT1, a founding member of the ZIP family[29–31]. So far, the underlying mechanisms of substrate specificity of the ZIPs are largely unknown. Note that although the exact metal species that is transported by ZIPs has not been fully established, $M^{2+}$ is used to indicate that the ZIPs transport divalent metal substrates.

In this work, we applied a systematic approach to rationally alter the substrate specificity of a multi-metal transporter, ZIP8 from human, initially aiming to increase the preference toward $Zn^{2+}$ over $Cd^{2+}$ (described Zn/Cd selectivity hereafter). The quadruple variant of ZIP8 created in this study (Q180H/E343H/C310A/N357H) exhibited drastically increased preference of $Zn^{2+}$ over not only $Cd^{2+}$ but also $Fe^{2+}$ and $Mn^{2+}$, which are the other two physiological substrates of ZIP8 besides $Zn^{2+}$. Structural modeling, evolutionary covariance analysis, and computational studies revealed a residue pair (Q180 and E343) that forms a selectivity filter at the entrance of the transport pathway only when ZIP8, an elevator-type transporter, adopts the outward-facing conformation, providing the structural and biochemical basis of the strong epistasis among the mutations introduced in the quadruple variant. The results of reverse substitution experiments performed on human ZIP4 confirmed the importance of the proposed selectivity filter in determining substrate preference.

## Results

**Identification of the differentially conserved residues (DCRs).** ZIP8 is a member of the LIV-1 subfamily including nine human ZIPs, in which ZIP8 and ZIP14 are multi-metal transporters mediating influx of $Zn^{2+}$, $Fe^{2+}$, $Mn^{2+}$, and $Cd^{2+}$ from the extracellular space[12,32]. The other LIV-1 proteins, including ZIP4, prefer $Zn^{2+}$ over other biologically relevant divalent metal ions. Accordingly, the LIV-1 subfamily can be divided into two groups with distinct substrate specificities. Multiple sequence alignment of human LIV-1 proteins allowed us to identify two residues, Q180 and E343 (residue numbers in ZIP8), which are invariable in ZIP8 and ZIP14 but replaced by histidine residues in the Zn-preferring LIV-1 proteins (Fig. 1a). We named the residues with these features differentially conserved residues (DCRs), as they are conserved in the orthologs (or close paralogs) sharing the same substrate specificity but non-conservatively replaced with the amino acids conserved in the paralogs with different substrate specificity. As it has been demonstrated that isozyme-specific residues are crucial in determining substrate specificity of enzymes[33–35], these two DCRs were postulated as key residues defining the substrate spectrum of the LIV-1 proteins. Notably, E343 is located in the transport site (the M1 site according to the BbZIP structures), whereas Q180 is predicted to be at the entrance of the transport pathway according to the ZIP8 structure model where the transporter adopts an inward-facing conformation (IFC) (Fig. 1b). In later experiments, we replaced Q180 and E343 with histidine and tested the substrate preference of the resulting variants. Sequence comparison also revealed two uniquely conserved metal-chelating residues (C374 and C376) in the putative M2 site of ZIP8, so we also tested the role of these residues by replacing them with the corresponding residues in ZIP4, a well-characterized zinc transporter[36–43].

**Internal competition transport assay to precisely measure the Zn/Cd selectivity.** Our initial focus was put on the Zn/Cd selectivity because the two metals, both in the IIB column of the periodic table, share similar properties in coordination chemistry. Indeed, the previous studies have indicated that $Zn^{2+}$ and $Cd^{2+}$ use the same transport pathway through multi-metal ZIPs[32,44], and distinguishing them requires a delicate but less understood mechanism. To precisely measure small kinetic isotope effect of an enzyme, a commonly used approach is the internal competition assay in which a mixture of two competing substrates labeled with light and heavy isotopes respectively is applied to the enzyme for processing[45]. Similarly, we applied an internal competition transport assay by adding a mixture of radioactive $^{65}Zn^{2+}$ and $^{109}Cd^{2+}$ to the cells transiently expressing the wild-type ZIP8 or the variants. After incubation and washing, radioactivity associated with the cells were measured to quantify $^{65}Zn$ and $^{109}Cd$ simultaneously by taking the advantage of different energy levels of the gamma rays emitted by the two radioisotopes ($^{65}Zn$: 800–1500 keV; $^{109}Cd$: 30–150 keV). Under the optimal experimental conditions (Supplementary Fig. 1), the calculated

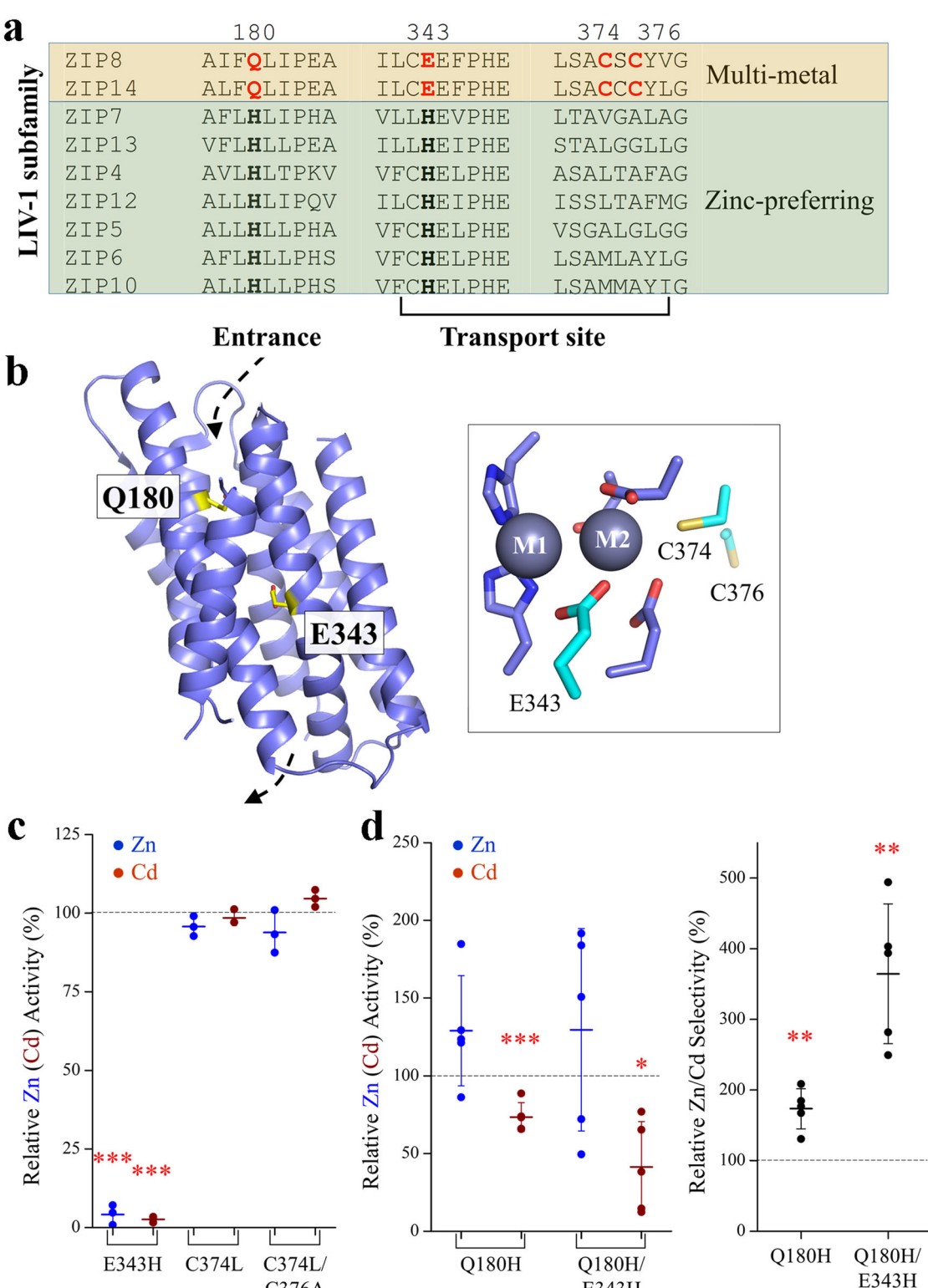

radioactivity ratio of $^{65}$Zn over $^{109}$Cd would represent the ratio of the transport rates of the two competing substrates (i.e., the Zn/Cd selectivity). An illustration of the internal competition transport assay is shown in Supplementary Fig. 2.

**Identification of a double variant with a greatly increased Zn/Cd selectivity.** The metal chelating residues in the transport site, including E343 in the M1 metal binding site as well as C374 and C376 in the M2 metal binding site[43,46], were tested in the first round. As shown in Fig. 1c, the mutations on the cysteine residues (C374L and C374L/C376A) yielded no change in Zn$^{2+}$ (or Cd$^{2+}$) transport activity, whereas the E343H variant exhibited no transport activity toward either substrate. This result was unexpected because the E343H mutation leads to a binuclear metal center with metal chelating residues identical to ZIP4, so it is unlikely that this mutation completely disrupts metal binding at the transport site. The abolished transport activity of the E343H variant could not be explained by misfolding either because the

**Fig. 1 Epistatic interaction between Q180 and E343 involved in determining transport activity and substrate preference of human ZIP8. a** Sequence alignment of human LIV-1 proteins to identify the uniquely conserved residues (in red) in the transport site and the pore entrance of ZIP8 and ZIP14. **b** The structure model of ZIP8 with the residues Q180 and H343 highlighted in stick mode. The dashed arrows indicate the entrance and the exit of the transport pathway. The transport site of ZIP8 with bound metal substrates (M1 and M2, gray spheres) is shown in the framed box, and the residues uniquely conserved in ZIP8 and ZIP14 are shown in cyan. **c** $Zn^{2+}$ and $Cd^{2+}$ transport activities of the variants with mutated transport site. The transport activities of the variants are expressed as percentages of the corresponding activities of the wild-type ZIP8. The $Zn^{2+}$ (or $Cd^{2+}$) transport activity was determined by subtracting the radioactivities of $^{65}Zn$ (or $^{109}Cd$) associated with the cells transfected with the empty vector from those associated with the cells transfected with the vector that contains the DNA encoding the wild-type ZIP8 or the variants. The shown data are from one representative experiment of two independent experiments with three replicates for each condition. The short horizontal bars represent ±S.D., and the long horizontal bars indicate means. *P* values (for those with significant differences): 0.0004, 0.0001. **d** The Q180H mutation rescued the transport dead E343H variant and increased the Zn/Cd selectivity. *Left*: The $Zn^{2+}$ and $Cd^{2+}$ transport activities of the variants of Q180H and Q180H/E343H. *Right*: The relative Zn/Cd selectivity of the variants of Q180H and Q180H/E343H. The Zn/Cd selectivity is defined as the ratio of the $Zn^{2+}$ transport activity over the $Cd^{2+}$ transport activity, and the relative Zn/Cd selectivity of a variant is expressed as percentage of the Zn/Cd selectivity of the wild-type ZIP8. Each solid dot represents the mean of at least three replicates in one individual experiment. The shown data are the combined results of five independent sets of experiments conducted for each variant. P values (for those with significant differences): 0.0031, 0.011, 0.0044, 0.004. The asterisks indicate the significant differences between the variants and the wild-type ZIP8 (Student's *t* tests: *$P \leq 0.05$; **$P \leq 0.01$; ***$P \leq 0.001$).

mutation did not cause a drastically reduced expression as indicated in Western blot (Supplementary Fig. 3a). We then tested the effects of the mutation of Q180 which is located at the entrance of the transport pathway. Of interest, the Q180H variant exhibited an increase in the Zn/Cd selectivity by nearly two folds when compared with the wild-type ZIP8 (Fig. 1d). More importantly, the Q180H mutation rescued the transport dead E343H variant, and the resulting Q180H/E343H double variant (the 2M variant hereafter) exhibited a nearly four-fold increase in the Zn/Cd selectivity relative to the wild-type ZIP8 and a twofold increase on top of the Q180H variant, indicative of strong synergistic effect between the two residues. Close inspection of the data showed that the increased Zn/Cd selectivity was achieved primarily through suppressing the $Cd^{2+}$ transport activity (Fig. 1d), whereas the $Zn^{2+}$ transport activity was modestly increased but with no statistical significance.

**Identification of additional mutations that further improve the Zn/Cd specificity.** By comparing the amino acid sequences among ZIP8, ZIP14, and ZIP4 homologs from multiple species (Fig. 2), many additional DCRs were identified. We chose ZIP4 to compare ZIP8 and ZIP14 because ZIP4 shares the highest sequence identity with ZIP8/14 than other LIV-1 members (Supplementary Table 1) and also it is a well-characterized zinc transporter. Mapping a total of 35 DCRs on the structure model of ZIP8 revealed a distinguishable pattern that many DCRs, particularly those potentially involved in metal chelation, are located along or at the entrance of the transport pathway (Figs. 2 and 3a). Based on the structure model, we postulate that, compared with the DCRs facing lipids or involved in TM packing, the metal chelating DCRs in the transport pathway are most likely to play a role in distinguishing metal substrates. We therefore examined whether replacing these DCRs in ZIP8 with the corresponding amino acids in ZIP4 would further increase the Zn/Cd selectivity on top of the 2M variant (Fig. 3b). The nine generated triple variants exhibited quite different properties: Some variants lost most (for the variants of 2M + N176D and 2M + E184K) or nearly all (for the variants of 2M + Q322T and 2M + P392E) of the transport activity toward both $Zn^{2+}$ and $Cd^{2+}$. The variant of 2M + D189H also exhibited reduced activity, but it can be partially attributed to the lower expression level according to the Western blot analysis. The variants of 2M + S325A and 2M + T451S showed no change in $Zn^{2+}$ or $Cd^{2+}$ transport when compared with the 2M variant, whereas the mutations of C310A and N357H seemed to increase the activity toward $Zn^{2+}$ more than $Cd^{2+}$. The total and cell surface expression analysis of the 2M and triple variants are shown in

Supplementary Fig. 3b–d. We then examined whether combining the C310A and N357H mutations with the 2M variant would improve the Zn/Cd selectivity. As shown in Fig. 3c, the resulting quadruple variant (Q180H/C310A/E343H/N357H, the 4M variant hereafter) exhibited a two times greater Zn/Cd selectivity than the 2M variant, which can be attributed to a significantly reduced $Cd^{2+}$ transport activity and a small (but statistically insignificant) decrease in $Zn^{2+}$ transport activity (Fig. 3c). When compared with the wild-type ZIP8, the 4M variant achieved an approximately 7–8 times enhancement in the Zn/Cd selectivity.

**Transport kinetic study of the 4M variant toward the multiple substrates of ZIP8.** Expression analysis showed that the 4M variant was expressed in HEK293T cells with similar total and cell surface expression levels when compared with the wild-type ZIP8 (Fig. 4a), indicating that the combined mutations did not affect folding or intracellular trafficking of the transporter. To understand the biochemical basis of the increased Zn/Cd selectivity, we measured $Zn^{2+}$ and $Cd^{2+}$ uptake and compared the kinetic parameters of the wild-type ZIP8 and the 4M variant. As shown in Fig. 4b, c, the 4M variant transported $Zn^{2+}$ with a nearly fourfold increase in $V_{max}$ but also a more than twofold increase in $K_M$, which led to a 1.6-times increase in $V_{max}/K_M$ when compared to the wild-type ZIP8. When transporting $Cd^{2+}$, the 4M variant exhibited a sixfold increase in $K_M$ with no significant change in $V_{max}$, resulting in a 6–7 times reduction in $V_{max}/K_M$. The cooperativity as indicated by the Hill coefficients ($n = 1$–$2$) may result from the M1 and M2 metal binding sites in the transport site and/or from the interactions between the two monomers of the ZIP dimer. Dimerization seems to be a common feature among the ZIP family members[9,11,38,47,48]. Together, the mutations in the 4M variant led to a moderate increase in specificity constant for $Zn^{2+}$ but a large reduction for $Cd^{2+}$, which accounted for the greatly enhanced Zn/Cd selectivity. In addition, the transport kinetics of the 4M variant was reminiscent of that of ZIP4 that transports $Zn^{2+}$ much more efficiently than $Cd^{2+}$ under the same experimental conditions (Supplementary Fig. 4). However, the residual $Cd^{2+}$ transport activity of the 4M variant suggests that there are additional unidentified mechanism(s) to allow the 4M variant to transport $Cd^{2+}$.

$Fe^{2+}$ and $Mn^{2+}$ are also natural substrates of ZIP8 and known to compete with other metals[19,21–23]. To examine whether the 4M variant exhibits altered transport activities toward these two metals, we compared the $Zn^{2+}$ transport activities of the wild-type ZIP8 and the 4M variant in the presence and absence of an excess amount of $Fe^{2+}$ or $Mn^{2+}$. As for the $Zn^{2+}/Fe^{2+}$ competition, $Zn^{2+}$ transport at the concentration of 5 μM significantly decreased for the wild-type ZIP8 when 100 μM $Fe^{2+}$ was added (Fig. 5a). In contrast, the 4M

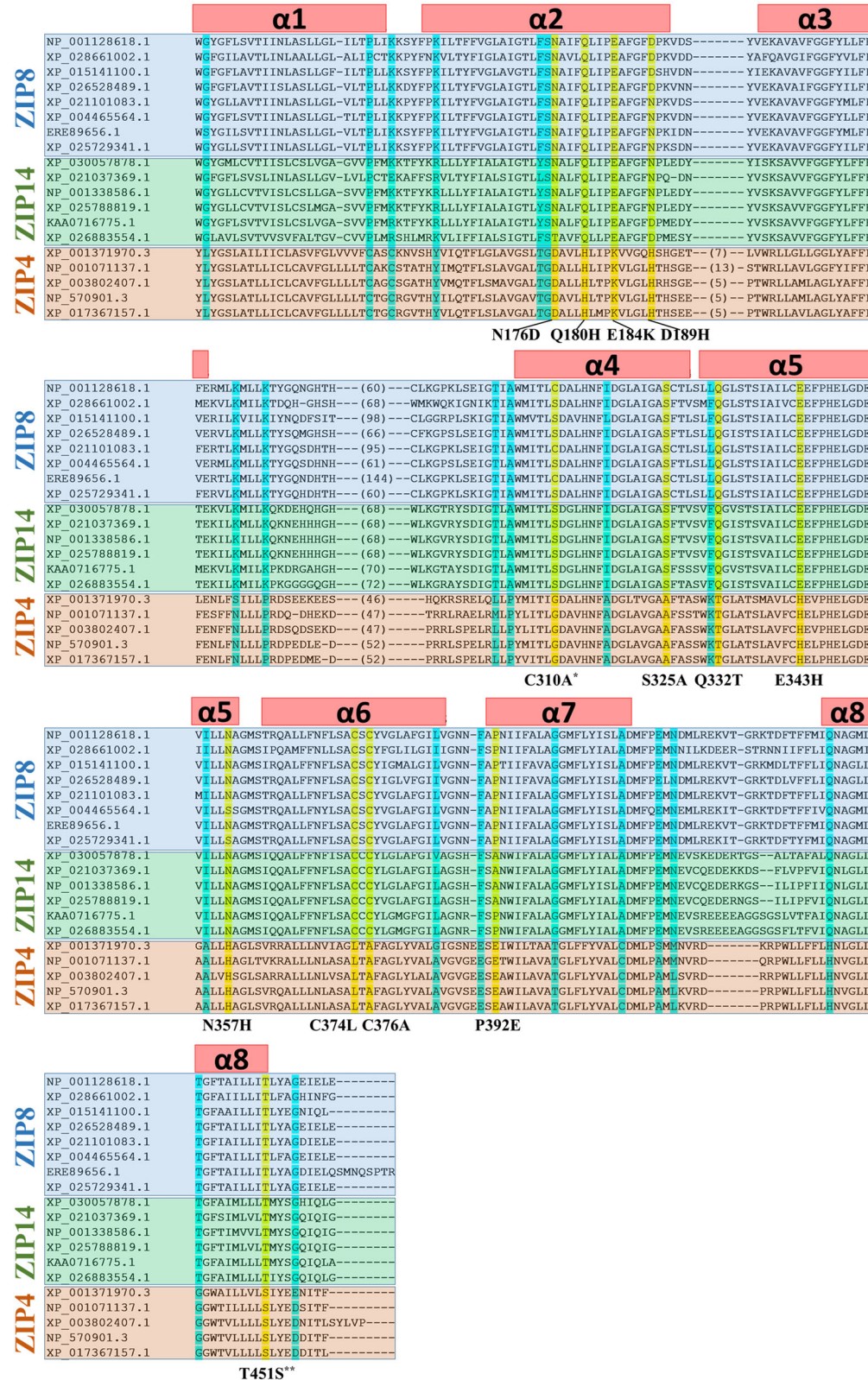

variant showed no sign of $Zn^{2+}$ transport inhibition by $100\,\mu M$ $Fe^{2+}$, suggesting that the 4M variant may have lost most of the transport activity toward $Fe^{2+}$. Consistently, the 4M variant showed a $V_{max}/K_M$ that is more than ten times lower than that for the wild-type ZIP8 with $V_{max}$ being reduced by seven folds and $K_M$ being increased by two folds (Fig. 5b). For $Mn^{2+}$, the presence of $Mn^{2+}$ at the concentration of $100\,\mu M$ efficiently blocked $Zn^{2+}$ transport of

the wild-type ZIP8 by more than 90% but had no inhibitory effect on the $Zn^{2+}$ transport activity of the 4M variant (Fig. 5c). Rather, an ~40% increase in $Zn^{2+}$ uptake was observed, indicative of a putative allosteric activation. Next, we measured the $Mn^{2+}$ transport activities of the wild-type ZIP8 and the 4M variant by using inductively coupled plasma mass spectrometry (ICP-MS). The Mn/phosphorus (Mn/P) molar ratio of the wild-type ZIP8 sample was significantly

**Fig. 2 Sequence alignment of the transmembrane domains of the paralogs of ZIP8 (with blue background), ZIP14 (green), and ZIP4 (pink).** ZIP8s: NP_001128618.1 (*Homo sapiens*), XP_028661002.1 (*Erpetoichthys calabaricus*), XP_015141100.1 (*Gallus gallus*), XP_026528489.1 (*Notechis scutatus*), XP_021101083.1 (*Heterocephalus glaber*), XP_004465564.1 (*Dasypus novemcinctus*), ERE89656.1 (*Cricetulus griseus*), XP_025729341.1 (*Callorhinus ursinus*). ZIP14s: XP_030057878.1 (*Microcaecilia unicolor*), XP_021037369.1 (*Mus caroli*), NP_001338586.1 (*Homo sapiens*), XP_025788819.1 (*Puma concolor*), KAA0716775.1 (*Triplophysa tibetana*), XP_026883554.1 (*Electrophorus electricus*). ZIP4s: XP_001371970.3 (*Monodelphis domestica*), NP_001071137.1 (*Rattus norvegicus*), XP_003802407.1 (*Otolemur garnettii*), NP_570901.3 (*Homo sapiens*), XP_017367157.1 (*Cebus imitator*). The DCRs along or at the entrance of the transport pathway are highlighted in yellow and the others are in cyan. The tested DCRs are potential metal chelating residues (before or after substitution) along the transport pathway or at the pore entrance. The mutations derived from DCR swapping are listed below the alignment. The transmembrane helices (α1-α8) are shown as the red bars above the sequences. *C310 was substituted with alanine to avoid potential structural disruption caused by glycine substitution. **T451S is a conservative replacement, but it was tested because T451 is spatially close to other selected DCRs.

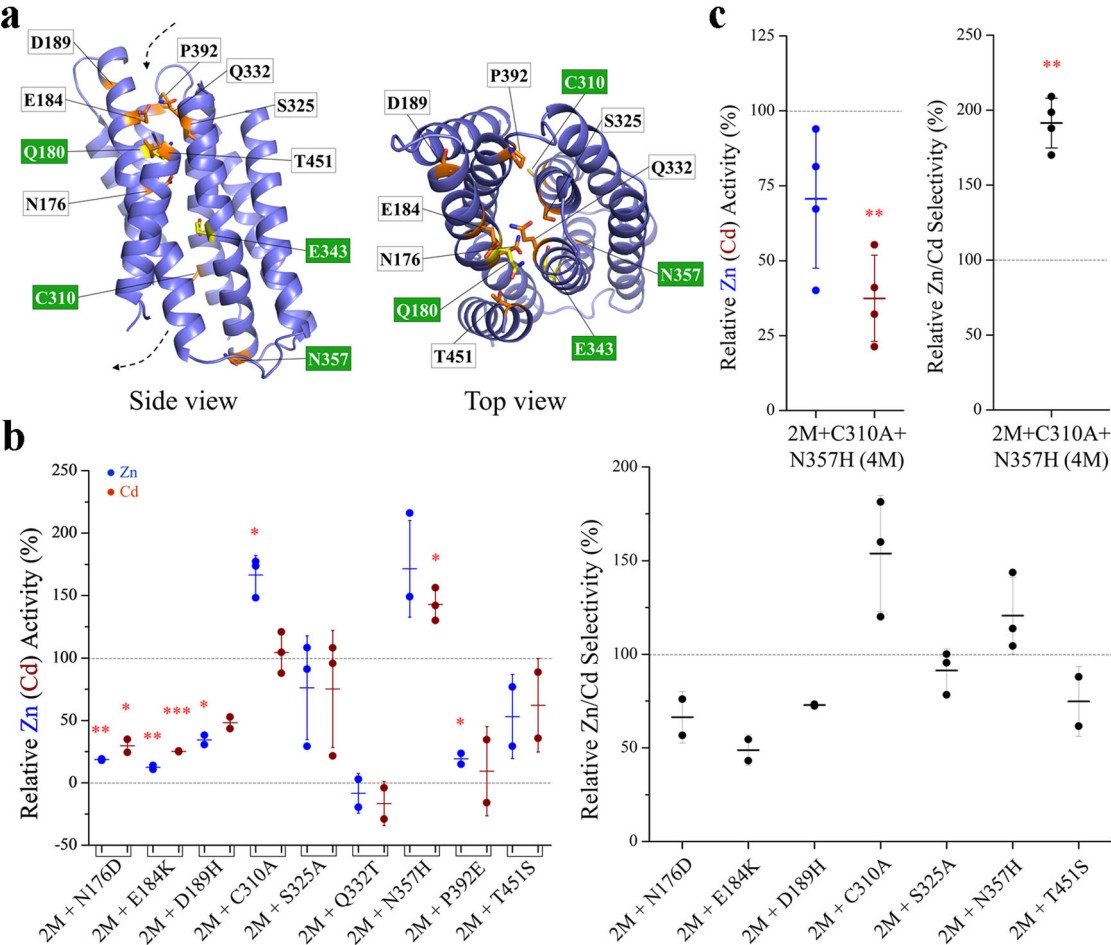

**Fig. 3 Generation of a quadruple variant with further improved Zn/Cd selectivity. a** Mapping of the selected DCRs on the structure model of ZIP8. The selected DCRs are labeled and shown in stick mode – Q180 and E343 are in yellow and the others are in orange. The residues substituted in the quadruple variant are labeled in white on a green background. **b** Screening of the triple variants with mutations on the selected DCRs. The Zn$^{2+}$ (Cd$^{2+}$) transport activities (left) and the Zn/Cd selectivity (right) are expressed as percentage relative to the Q180H/E343H double variant (2M). Each solid dot represents the mean of at least three replicates in one individual experiment. The shown data are the combined results of 2–3 independent sets of experiments conducted for each variant. The short horizontal bars represent S.D., and the long horizontal bars indicate means. *P* values (for those with significant differences): 0.0046, 0.047, 0.012, 0.0025, 0.036, 0.018, 0.03, 0.034. **c** Comparison of the Zn (Cd) transport activity (left) and the Zn/Cd selectivity (right) of the quadruple variant (4M) with those of the 2M variant. The Zn (Cd) transport activities (left) and the Zn/Cd selectivity (right) are expressed as the percentage values relative to the 2M variant. Each data point represents the mean of three replicates in one experiment and the results of four independent sets of experiments are shown. *P* values (for those with significant differences): 0.0032, 0.0016. The asterisks indicate the significant differences between the 2M and the triple variants or between the triple and the quadruple (4M) variants (Student's *t* tests: *$P \leq 0.05$; **$P \leq 0.01$; ***$P \leq 0.001$).

increased, whereas no change in the Mn/P ratio was observed for the 4M variant sample when compared with the control group (Fig. 5d). These results strongly indicated that the 4M variant has lost the Mn$^{2+}$ transport activity.

Collectively, by substituting four DCRs of the multi-metal transporter ZIP8 with the amino acids at the corresponding positions of the Zn transporter ZIP4, we were able to create a ZIP8 variant with a drastically increased substrate preference toward

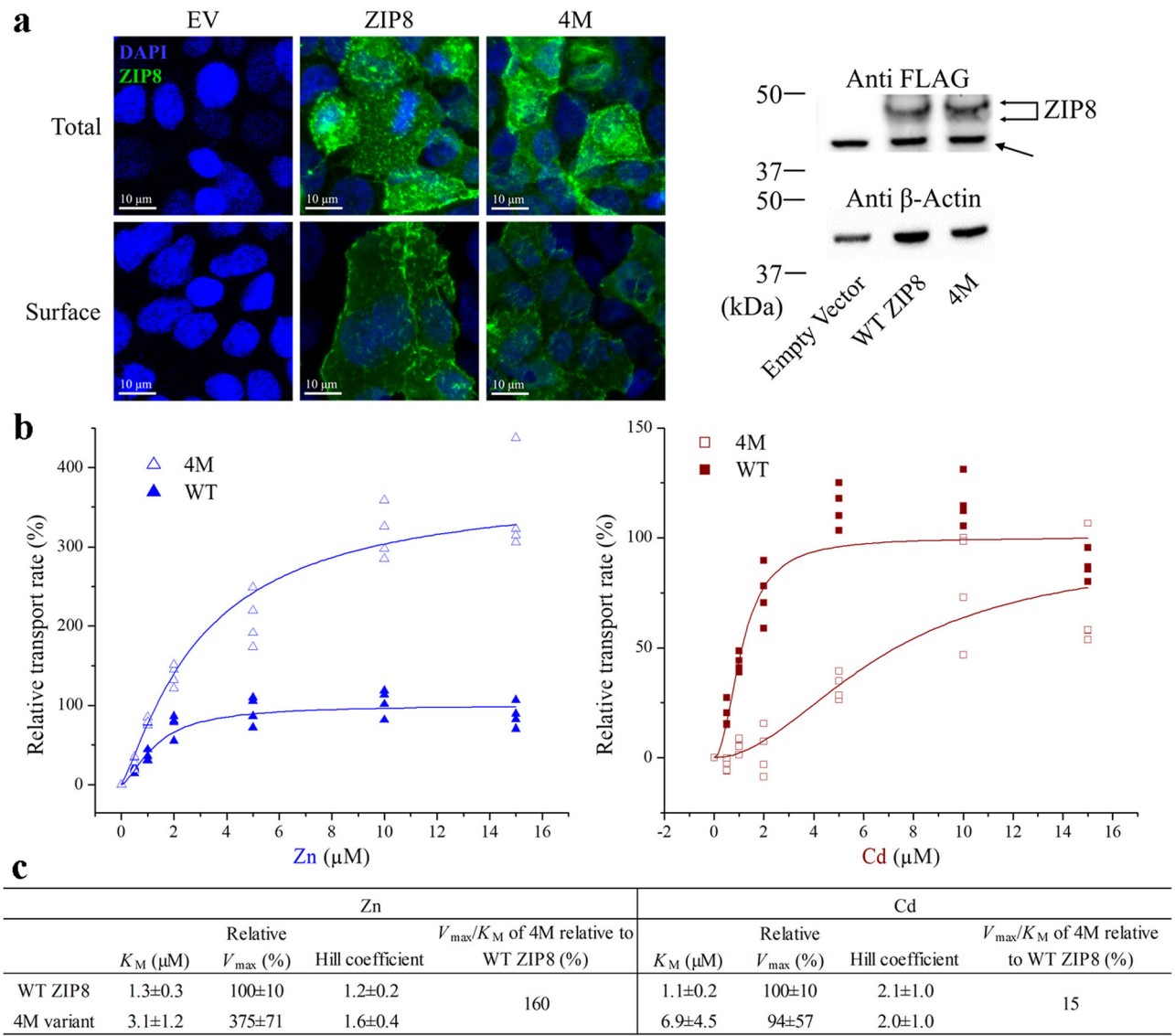

**Fig. 4 Transport kinetics of the 4M variant in comparison with that of the wild-type ZIP8. a** Expression analysis of the wild-type ZIP8 and the 4M variant in HEK293T cells. Left: Immunofluorescence images of the cells expressing the wild-type ZIP8 or the 4M variant. The total expression and the surface expression of the N-FLAG ZIP8 (or the 4M variant) were detected by an anti-FLAG antibody and then an Alexa-488 labeled secondary antibody in permeabilized and non-permeabilized cells, respectively. The scale bar represents 10 μm. Right: Western blot of the wild-type ZIP8 and the 4M variant. β-Actin was used as loading control. A non-specific signal is indicated by the arrow. **b** Transport of $Zn^{2+}$ (left) and $Cd^{2+}$ (right) by the 4M variant (open symbols) and the wild-type ZIP8 (solid symbols). The shown data are from one representative experiment of three independent experiments with 3-4 replicates for each data point. The curves were fitted using the Hill model in Origin$^{TM}$. The transport rates are expressed as percentages of the $V_{max}$ value of the wild-type ZIP8. **c** The kinetic parameters of the 4M variant and the wild-type ZIP8. The errors are the S.E. generated in curve fitting.

$Zn^{2+}$ over $Cd^{2+}$, $Fe^{2+}$, and $Mn^{2+}$ due to the enhanced $Zn^{2+}$ transport and the greatly diminished transport for the other metals.

**A conditional selectivity filter that is formed and functions only in the outward-facing conformation (OFC).** In order to understand the structural and biochemical basis of the strong epistatic interaction between Q180 and E343 (Fig. 3), we conducted structural modeling, evolutionary covariance analysis, and phylogenetic analysis, which revealed a conditional selectivity filter that functions only in the OFC.

Using the IFC structure and the OFC structure model of BbZIP as templates, we generated the homology models of human ZIP8 in two conformational states (Fig. 6a). We noticed that AlphaFold predicts the structures of human ZIPs in distinct conformations with several ZIPs being in the OFC state[11]. Using the predicted

structure of ZIP13, which represents the most outward open state, as the template, we generated an additional ZIP8 structural model in the OFC (Fig. 6a). Remarkably, although the $C_\alpha$ atoms of Q180 and E343 are 14.9 Å apart in the IFC model, they approach one another in both OFC models where they are 7.5 Å and 8.9 Å apart, respectively. Consistently, Q180 and E343 are predicted to form a direct interaction according to an evolutionary covariance analysis of ZIP8 (Fig. 6b). As these metal chelating residues are close in space and face each other, they likely form a metal binding site at the entrance of the transport pathway.

Next, we conducted a phylogenetic analysis of the Q180-E343 pair in the entire ZIP family. Sequence analysis of more than 11,000 ZIPs from the major subfamilies of the ZIP revealed that the topologically equivalent residue of E343 of ZIP8 is histidine in most of the subfamilies, while glutamate occupies this position

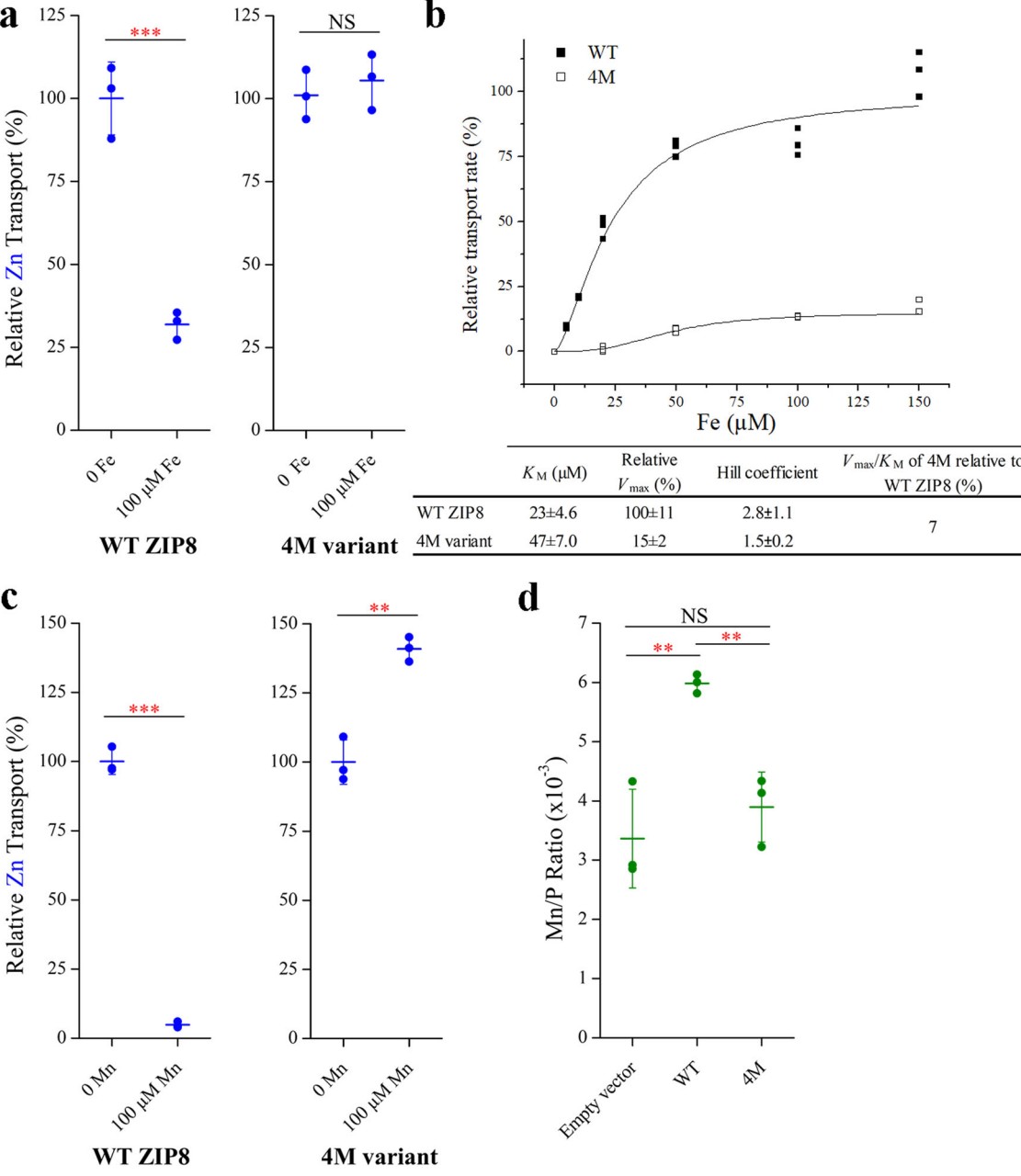

**Fig. 5 Reduced Fe$^{2+}$ and Mn$^{2+}$ transport activity of the 4M variant. a** Inhibition of Zn$^{2+}$ transport of the wild-type ZIP8 and the 4M variant by 100 μM of Fe$^{2+}$. The shown data are from one representative experiment of two independent experiments with three replicates for each condition. *P* value: 0.00055. **b** Fe$^{2+}$ transport kinetics of the wild-type ZIP8 and the 4M variant. The shown data are from one representative experiment of three independent experiments with three replicates for each condition. The curves were fitted using the Hill model in Origin$^{TM}$. The transport rates are expressed as percentages of the $V_{max}$ of the wild-type ZIP8. The kinetic parameters of the 4M variant and the wild-type ZIP8 are listed in the table and the errors are the S.E. generated in curve fitting. **c** Inhibition of Zn$^{2+}$ transport of the wild-type ZIP8 and the 4M variant by 100 μM of Mn$^{2+}$. The shown data are from one representative experiment of three independent experiments with three replicates for each condition. *P* values: 4.2E−06, 0.0015. **d** Mn$^{2+}$ transport assay of the wild-type ZIP8 and the 4M variant. After incubation with Mn$^{2+}$ (50 μM) for 30 min, the cells were washed and digested by nitric acid for ICP-MS measurement. The intracellular Mn levels were expressed as the molar ratios of Mn and phosphorus in the same sample. The shown data are from one representative experiment of two independent experiments with three replicates for each condition. *P* values (for those with significant differences): 0.0059, 0.0041. In (**a**, **c**, **d**), the short horizontal bars represent S.D., and the long horizontal bars indicate means. The asterisks indicate the significant differences (Student's *t* tests: **$P ≤ 0.01$; ***$P ≤ 0.001$; NS, not significant).

only in a small fraction of the eukaryotic LIV-1 proteins (Fig. 6c). At the position equivalent to Q180 of ZIP8, a glutamine, which is exclusively present in the eukaryotic members of the LIV-1 subfamily, always copresents with a glutamate at the position equivalent to E343 of ZIP8, suggesting that the functions of the two residues are strongly coupled, which is consistent with the strong epistasis as revealed in the mutagenesis study (Fig. 3).

Taken together, given the short distance between Q180 and E343 in the OFC models and their functional coupling in determining substrate preference and activity, Q180 and E343 are

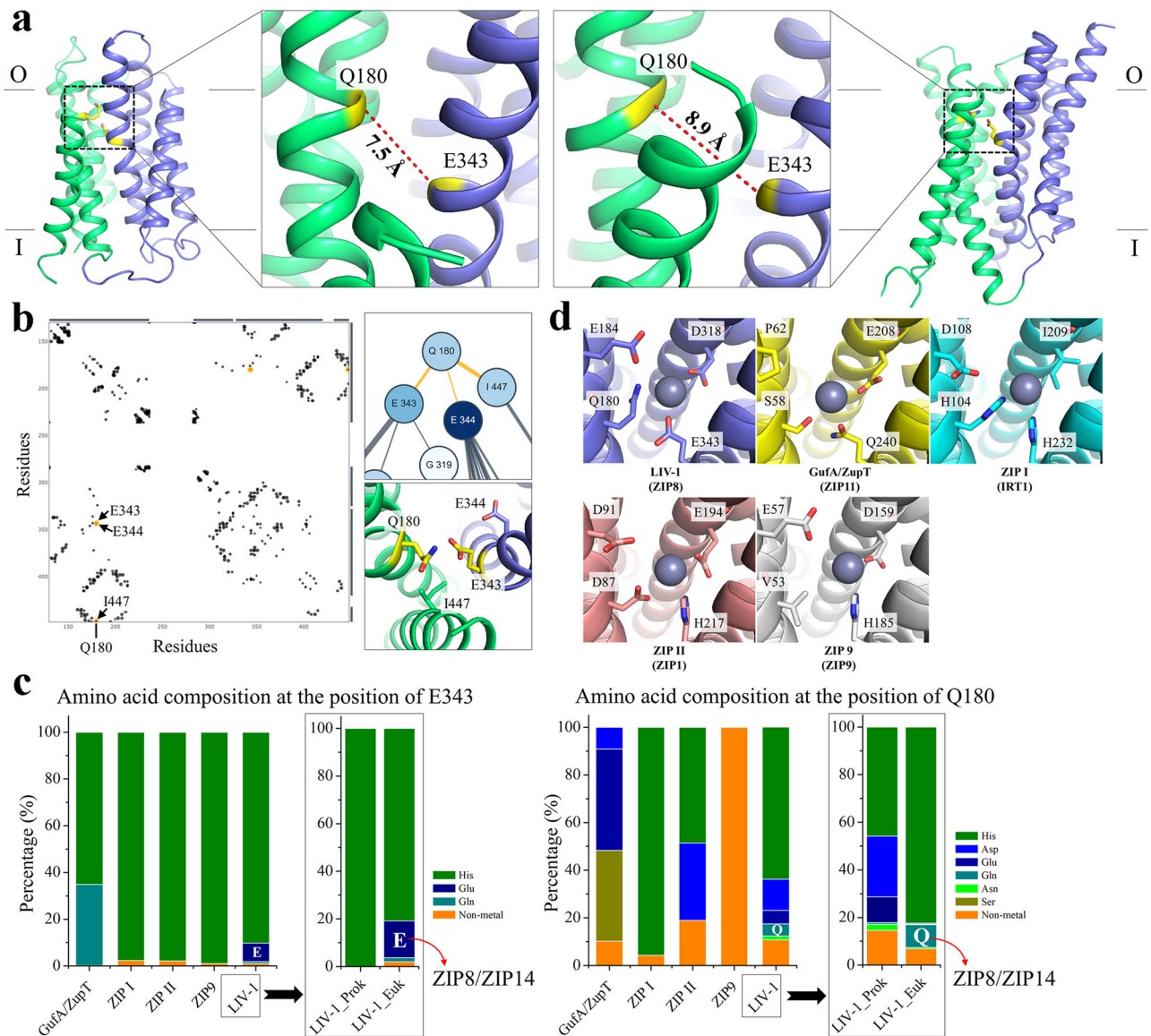

**Fig. 6 Identification of a conditional selectivity filter at the entrance of the transport pathway. a** ZIP8 structural models in the OFC. The structural models were generated using either the OFC model of BbZIP (left) or the AlphaFold predicted human ZIP13 (right) as the template in homology modeling. As shown in the zoomed-in views, Q180 and E343 are close and face to each other in both structural models with the distances between the C$_\alpha$s being labeled. The scaffold domain is colored in green and the transport domain in blue. **b** Predicted contacting residues of ZIP8 in an evolutionary covariance analysis by EVcouplings. Interactions involving Q180 are highlighted in orange in the interaction diagram (left). E343, E344, and I447 are predicted to interact with Q180, which is consistent with the structural model of ZIP8 in the OFC (right). **c** Amino acid composition at the positions of E343 (left) and Q180 (right) in major ZIP subfamilies. Analysis of prokaryotic and eukaryotic ZIPs in the LIV-1 subfamily is shown in the frames. The amino acid sequences retrieved from the PF02535 family in the Pfam database, including 2599 in GufA/ZupT subfamily, 1200 in ZIP I subfamily, 716 in ZIP II subfamily, 5955 in LIV-1 subfamily, and 1033 in ZIP9 subfamily, were aligned and analyzed in Jalview. **d** The selectivity filter of representative ZIPs (in parentheses) from the major subfamilies. The structural models were generated using the AlphaFold predicted human ZIP13 as the template in homology modeling. The residues potentially involved in metal binding in ZIP8 and the corresponding residues in other ZIPs are labeled and shown in stick mode. Some side chains were adjusted to better accommodate a zinc ion (gray sphere).

likely involved in a selectivity filter at the entrance of the transport pathway. As Q180 and E343 approach only when the transporter adopts an OFC, we call it a conditional selectivity filter. Close inspection of the OFC models suggested that D318 from the transport site may also join Q180 and E343 to better coordinate metal ions at the pore entrance (Fig. 6d) and the nearby E184 is another candidate involved in metal recruitment. Indeed, the E184K mutation reduced the activity of the 2M variant by 80–90% (Fig. 3). Sequence analysis of the entire ZIP family and modeling studies suggested that this conditional

selectivity filter may be present in most of the ZIP family members, although the amino acid composition of this metal binding site may vary in different subfamilies (Fig. 6d).

**Free energy calculation of metal binding to the selectivity filter.** Substitution of Q180 and E343 by histidine residues introduced two imidazole groups in the selectivity filter (Fig. 7a), generating a metal binding site preferring $Zn^{2+}$ over $Fe^{2+}$ and $Mn^{2+}$ according to the previously determined free energies of metal binding with

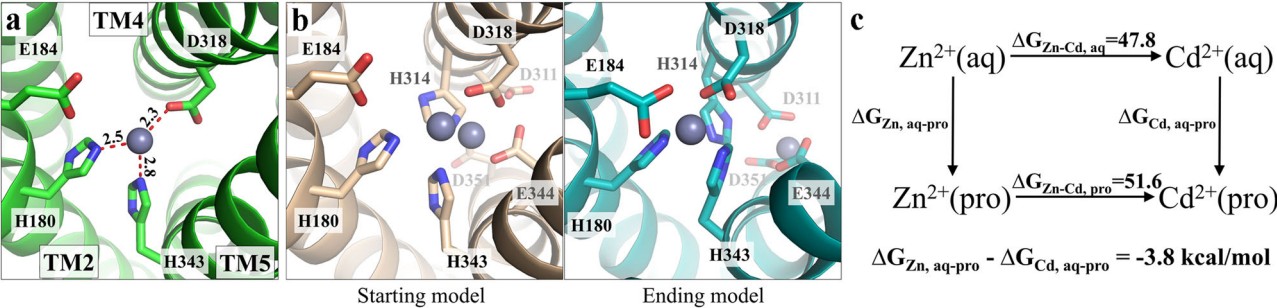

**Fig. 7 Computational characterization of the interactions of the 4M variant with metal substrates at the selectivity filter. a** Structural model of the 4M variant with a zinc ion bound (gray sphere) at the selectivity filter. The structural model was generated by homology modeling using the human ZIP13 structure predicted by AlphaFold as the template. The metal ion is coordinated with H180, D318, and H343 with the metal-ligand distances labeled in Angstrom. **b** MD simulations of the 4M variant with two bound zinc ions. Left: the starting model. One zinc ion was placed at the selectivity filter and the other at the transport site to prevent the former from entering the channel. Right: the ending model. The metal ion initially placed at the selectivity filter was stabilized by four residues, including E184, whereas the second one in the transport site moved down the transport pathway to a deeper position where two aspartate residues (D311 and D351, highly conserved in the LIV-1 subfamily) joined E344 to form a new metal binding site. **c** Free energy calculation of $Zn^{2+}$ and $Cd^{2+}$ binding to the selectivity filter (with E184 included) of the 4M variant. $M^{2+}$(aq) and $M^{2+}$(pro) indicate $Zn^{2+}$ (or $Cd^{2+}$) in aqueous solution or bound at the selectivity filter, respectively. The free energy change resulting from the mutation of $Zn^{2+}$ to $Cd^{2+}$ in aqueous solution ($\Delta G_{Zn-Cd, aq}$) or at the selectivity filter ($\Delta G_{Zn-Cd, pro}$) was calculated using thermodynamic integration and the results are expressed in kcal/mol.

small molecule ligands (Supplementary Fig. 5a)[49,50]. However, this estimation cannot satisfactorily explain why histidine replacement also improved the Zn/Cd selectivity since the free energy reduction upon ligand replacement with imidazole is similar for $Zn^{2+}$ and $Cd^{2+}$. To address this issue, we conducted molecular dynamics (MD) simulations on the 4M variant in the OFC with a bound metal ion at the selectivity filter to obtain an energy-minimized structural model, which will be subsequently used for free energy calculation of metal binding using thermodynamic integration[49–55]. In the first trial, only after 10 ns of MD simulation, the $Zn^{2+}$ initially located at the selectivity filter moved to a deeper place within the transport pathway where the metal ion was coordinated by the residues from both M1 and M2 sites (Supplementary Fig. 5b). To prevent metal ion translocation to the transport site, a second metal ion was added in the starting model at the position which would otherwise be occupied by the metal ion at the selectivity filter. As expected, the metal ion bound at the selectivity filter was stabilized throughout a 3 μs MD simulation only with a small displacement (Fig. 7b). Notably, E184, which was not included in metal chelation in the starting model, was found to join the metal binding site with H180, H343, and D318 during the MD simulations, likely improving metal binding at the selectivity filter. Meanwhile, due to the repulsion between the two metal ions in the starting model, the second metal ion added in the transport site moved further down the transport pathway and then stabilized by three carboxylic acid residues (E344, D311, and D351). Using this energy-minimized and structure-stabilized ending model, we calculated and compared the free energy changes of $Zn^{2+}$ and $Cd^{2+}$ binding at the selectivity filter. As shown in Fig. 7c, the selectivity filter (with E184 being included) of the 4M variant prefers binding $Zn^{2+}$ over $Cd^{2+}$ by 3.8 ± 0.3 kcal/mol ($n = 3$). This calculated free energy difference reflects the apparent $K_M$ difference to some degree, but may be overestimated due to theoretical and experimental factors, including the accuracy of the simulation model, as well as not taking into account the rates of protein conformational change and metal release from the transport site, which affect $K_M$. Nevertheless, the results of the free energy calculation supported that the amino acid composition of the identified selectivity filter critically determines the substrate preference and that metal screening at the selectivity filter is a crucial step of distinguishing metal substrates.

**Characterization of the ZIP4 variants containing the reverse substitutions**. To test the importance of the selectivity filter in another ZIP, we chose human ZIP4 to perform reverse substitution on the residues that are topologically equivalent to those that were mutated in the 4M variant of ZIP8. As shown in Fig. 8a, for $Zn^{2+}$ and $Cd^{2+}$ transport, the results indicated that: (1) The H379Q mutation (the reverse substitution of Q180H in ZIP8) showed little effect on $Zn^{2+}$ or $Cd^{2+}$ transport activity; (2) The H536E mutation (the reverse substitution of E343H in ZIP8) completely abolished the $Zn^{2+}$ transport activity but retained a marginal activity toward $Cd^{2+}$; (3) Combining the H379Q and H536E mutations (the reverse substitutions of the 2M variant) partially restored $Zn^{2+}$ and $Cd^{2+}$ transport activities and the Zn/Cd selectivity was significantly reduced by more than 60%, mirroring the effect of the 2M variant of ZIP8 that exhibits an increased Zn/Cd selectivity; and (4) Incorporation of additional two mutations (G503C and H550N, the additional two reverse substitutions of the 4M variant) with the H379Q/H536E variant abrogated $Zn^{2+}$ and $Cd^{2+}$ transport activities. Importantly, the partial restoration of $Zn^{2+}$ and $Cd^{2+}$ activities of the H536E variant by the H379Q mutation suggests an epistatic interaction, consistent with the proposed structural model where residues at these two positions are in close proximity when the transporter is in the OFC (Fig. 6). The significantly reduced Zn/Cd selectivity of the H379Q/H536E variant reinforces the notion that these two residues at the selectivity filter play a role in determining substrate specificity. As the H379Q mutation had little effect on $Zn^{2+}$ or $Cd^{2+}$ transport activity, it seems that the H536E mutation is responsible for the reduced Zn/Cd selectivity. Indeed, the single H536E mutation did not completely abrogate $Cd^{2+}$ transport activity as it did for $Zn^{2+}$ transport. While we could not detect $Fe^{2+}$ transport activity for any of the tested constructs of ZIP4, a $Mn^{2+}$ transport activity was detected for the wild-type ZIP4 and the H379Q variant (Fig. 8b, c). The $Mn^{2+}$ transport activity was diminished by the H536E mutation but could not be restored by the H379Q mutation. Interestingly, the quadruple variant (H379Q/H536E/G503C/H550N) exhibited a partially restored $Mn^{2+}$ activity. Overall, the mutagenesis study on ZIP4 confirmed the importance of the selectivity filter in determining substrate preference, whereas the other two mutations along the transport pathway appeared to function differently from their counterparts in ZIP8.

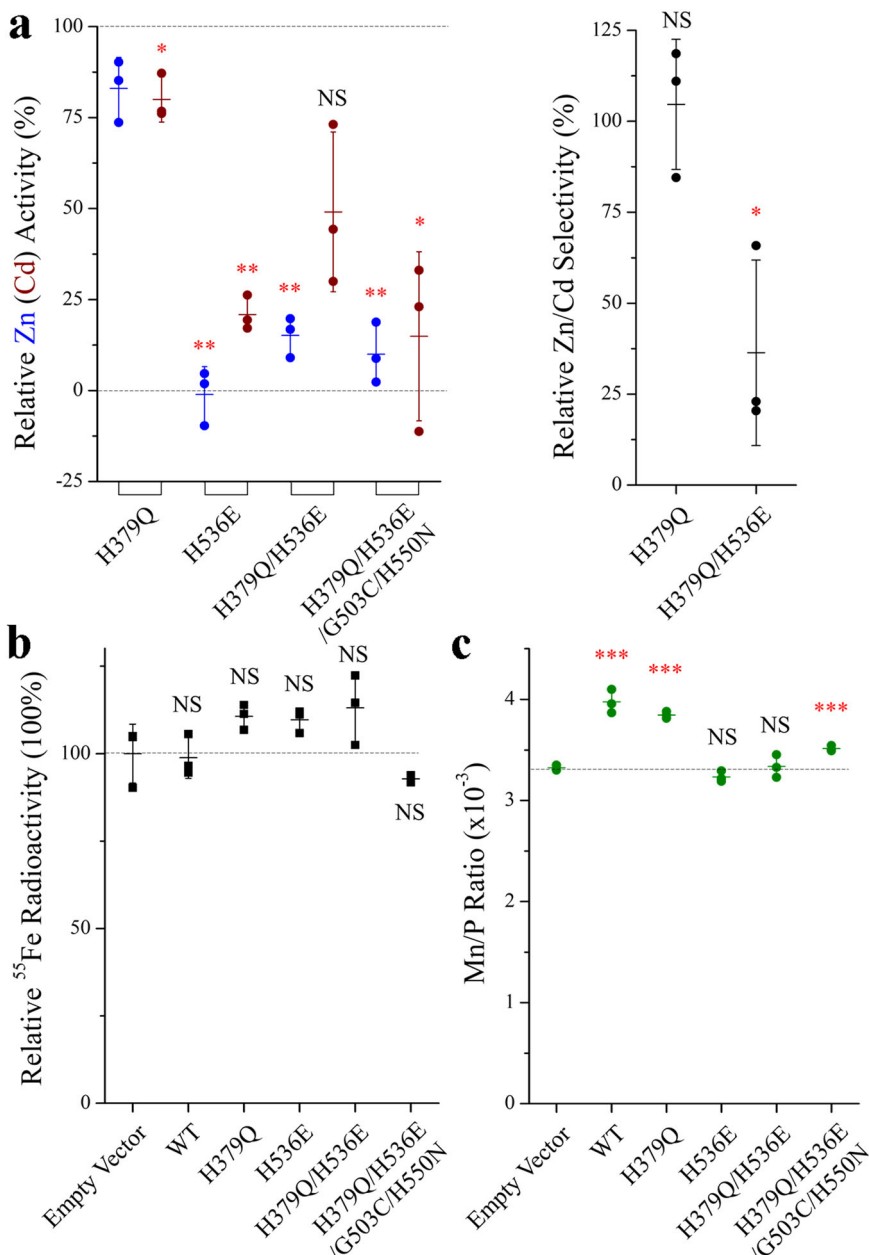

**Fig. 8 Characterization of the substrate specificities of human ZIP4 and its variants. a** The $Zn^{2+}$ ($Cd^{2+}$) transport activities (left) and the Zn/Cd selectivity (right) are expressed as percentage relative to the wild-type ZIP4. Each solid dot represents the mean of three replicates in one individual experiment. The shown data are the combined results of three independent sets of experiments conducted for each variant. The short horizontal bars represent S.D., and the long horizontal bars indicate means. $P$ values (for those with significant differences): 0.074, 0.031, 0.0019, 0.0012, 0.0014, 0.0028, 0.024, 0.05. **b** $Fe^{2+}$ transport assay of the wild-type ZIP4 and the variants. The $^{55}Fe$ radioactivity associated with the cells transfected with the empty vector was set as 100%. The shown data are from one representative experiment of three independent experiments with three replicates for each condition. **c** $Mn^{2+}$ transport assay of the wild-type ZIP4 and the variants. After incubation with $Mn^{2+}$ (50 μM) for 30 min, the cells were washed and digested by nitric acid for ICP-MS measurement. The intracellular Mn levels were expressed as the molar ratios of Mn and phosphorus in the same sample. The shown data are from one representative experiment of three independent experiments with three replicates for each condition. The asterisks indicate the significant differences between the wild-type ZIP4 and the variants (for Zn and Cd) or between the empty vector group and the transporter groups (for Fe and Mn). $P$ values (for those with significant differences): 0.00067, 3.1E−05, 0.0009. Student's $t$ tests: $*P \leq 0.05$; $**P \leq 0.01$; $***P \leq 0.001$. The expression of the HA-tagged ZIP4 and the variants were analyzed by Western blots (Supplementary Fig. 6).

## Discussion

Engineering of metal transporters with altered substrate specificity can be strategized for potential applications in agriculture and environmental protection. In this work, we developed and applied an approach to rationally alter the substrate specificity of a multi-metal ZIP transporter, human ZIP8. We created a zinc-preferring ZIP8 variant by combining four designed mutations on

selected residues at the entrance or along the transport pathway, demonstrating that these mutations increased the $Zn^{2+}$ transport activity while largely suppressed the activities toward $Cd^{2+}$, $Fe^{2+}$, and $Mn^{2+}$ (Figs. 1–4). To apply this approach on a transporter of interest, it is crucial to identify a close homolog (referred to as the reference) with a distinct substrate spectrum. ZIP4 was chosen as the reference of ZIP8 because it strongly prefers $Zn^{2+}$ over $Cd^{2+}$

(Supplementary Fig. 4)[36,43,56]. One can identify the DCRs through multiple sequence alignment of the homologs of the target and the reference, map them on the structure (or structure model) of the transporter, and find specific residues located at the entrance or along the transport pathway. Then, the selected DCRs are replaced with the amino acids in the corresponding positions of the reference transporter. The resulting variants are subjected to internal competition transport assay to identify the mutations which confer a desired substrate specificity to the variants. Finally, the beneficial mutations can be combined for greater fitness improvement. This approach is likely applicable to other transporters and it can also be combined with directed evolution. For instance, optimization of the amino acids at the key residues identified from the DCR swapping study can be pursued by multi-site saturation mutagenesis.

More importantly, the mutagenesis and functional studies conducted in this work unraveled key molecular determinants of substrate specificity of ZIP8, in particular a conditional selectivity filter. Among the four residues substituted in the 4M variant, only E343 participates in metal binding at the M1 site which is the primary transport site of the ZIPs[6,44,46,57]. Previous multiple sequence analysis has predicted E343 as a key residue involved in determining substrate specificity[14,15], but its role was evidenced only when the transport dead E343H variant was rescued by the Q180H mutation in the 2M variant (Fig. 1). The importance of the equivalent residue of Q180 in other ZIPs, including H104 in IRT1[6], H379 in ZIP4[46,48], and S106 in BbZIP[9], were documented, and it has been proposed that the residue at this position is at the very frontline in interacting with metal substrates[46,48]. Consistently, this work finds that Q180 in ZIP8 plays a pivotal role in determining substrate preference. Structural modeling, covariance analysis, phylogenetic analysis, and free energy calculation provided the structural and physicochemical basis of the epistatic interaction and synergy of Q180 with E343, strongly indicating that Q180 and E343 not only jointly exert functions in selecting substrates but also physically interact at the entrance of the transport pathway in the OFC (Figs. 5 and 6). Physical contact of the involved residue pair is one of many reasons that explain the epistasis of two compensatory mutations[58]. The importance of this residue pair in determining substrate preference and a similar epistatic interaction between these two residues have been confirmed through the corresponding reverse substitutions in ZIP4 (Fig. 8), a zinc transporter in the same LIV-1 subfamily. Compared with other known elevator transporters, ZIP8 (and possibly many other ZIPs as well as) is unique in that Q180 in the scaffold domain approaches E343 in the transport domain to screen metal substrates (Fig. 6a). This is a prominent example showing that the scaffold domain of an elevator transporter may actively participate in substrate selection. Different from the pre-formed selectivity filters in ion channels transporting alkaline and alkaline earth metal ions, the selectivity filter of ZIP8, which is a likely carrier according to the saturable kinetic profile (Figs. 4 and 5), is formed in a conformation dependent manner, which therefore represents an unprecedented conditional selectivity filter. Also unlike the selectivity filters of canonical ion channels, the proposed selectivity filter of ZIP8 must not be the only mechanism that determines substrate specificity, since residues in the transport site and along the transport pathway (see below) are the obvious candidates involved in discriminating metal substrates.

The other two mutations in the 4M variant (C310A and N357H) are distant from the transport site or the entrance of the transport pathway. Rather, they are located at the interface between the transport domain and the scaffold domain. Although each mutation had very modest effects, combination of the two mutations significantly improved the Zn/Cd selectivity on top of the 2M double variant (Figs. 2 and 3). One possible explanation is

that these mutations at the domain interface alter the dynamics of the elevator-type transporter, i.e., the sliding of the transport domain against the scaffold domain. It has been shown that the mutations at the domain interface of a prototypic elevator transporter UapA drastically changed the substrate spectrum[59]. In addition, mutations of metal chelating residues may also alter the dynamic interactions of metal substrate with the transporter along the transport pathway. This is consistent with the notion that substrate translocation dynamics is as crucial as modification of the transport site in defining the substrate spectrum of a sugar transporter[60]. Accordingly, the entire process of multiple dynamic interactions between substrate molecule and transporter along and at the entrance of the conduit, rather than the mere substrate binding at the high-affinity transport site, is vital for substrate recognition and selection. Uniquely for metals, their interactions with a variety of ligands in solution (water, counterions, and other small molecule ligands) should also be considered, as the ligand exchange rates of metal-ligand complexes can affect metal selectivity as well.

Taken together, we demonstrated in this work that substitution of four rationally selected residues was sufficient to convert human ZIP8 that transports multiple metal substrates into a zinc-preferring transporter. Importantly, a conditional selectivity filter that is formed and functions only when the transporter adopts the OFC was identified by using combined approaches. As the 4M variant is a zinc-preferring transporter with similar total and cell surface expression as the wild-type ZIP8 (Fig. 4a), it can be applied to knock-out/knock-in experiments to determine whether certain biological functions of ZIP8 are attributed to the transport activity toward $Zn^{2+}$ and/or other natural substrates ($Fe^{2+}$ and $Mn^{2+}$). Given the high sequence identity between ZIP8 and ZIP14 (48%, Supplementary Table 1), the same mutations in ZIP14 may generate similar effects on substrate specificity. After all, the importance of the proposed selectivity filter has been confirmed in ZIP4 (Fig. 8), which shares a lower sequence identity (31%, Supplementary Table 1). Whether or not the identified residue pair plays a similar role in ZIPs with longer evolutionary distance needs to be tested in future study.

## Methods

**Genes, plasmids, and mutagenesis.** The complementary DNA of human ZIP4 (GenBank access number: BC062625) and human ZIP8 (GenBank access number: BC012125) from Mammalian Gene Collection were purchased from GE Healthcare. The ZIP4 coding sequence was inserted into a modified pEGFP-N1 vector (Clontech) in which the downstream EGFP gene was deleted and an HA tag was added at the C-terminus. The ZIP8 coding sequence was inserted into the pcDNA3.1 vector (Invitrogen). The ZIP8 construct consists of the N-terminal signal peptide of ZIP4 (amino acid residues 1–22) followed by a GSGS linker and a FLAG tag, the ZIP8 coding sequence (residue 23–460), and a HA-tag at the C-terminus. Site-directed mutagenesis of ZIP8 was conducted using QuikChange mutagenesis kit (Agilent, Cat#600250). All mutations were verified by DNA sequencing. The primers used in this work are listed in Supplementary Table 2.

**Mammalian cell culture, transfection, and Western blot.** Human embryonic kidney cells (HEK293T, ATCC, Cat#CRL-3216) were cultured in Dulbecco's modified eagle medium (DMEM, Thermo Fisher Scientific, Invitrogen, Cat#11965092) supplemented with 10% (v/v) fetal bovine serum (FBS, Thermo Fisher Scientific, Invitrogen, Cat#10082147) and Antibiotic-Antimycotic solution (Thermo Fisher Scientific, Invitrogen, Cat# 15240062) at 5% $CO_2$ and 37 °C. Cells were seeded on the polystyrene 24-well trays (Alkali Scientific, Cat#TPN1024) for 16 h in the basal medium and transfected with 0.8 μg DNA/well using lipofectamine 2000 (Thermo Fisher Scientific, Invitrogen, Cat# 11668019) in DMEM with 10% FBS.

For Western blot, samples were mixed with the SDS sample loading buffer and heated at 96 °C for 10 min before loading on SDS-PAGE gel. The proteins separated by SDS-PAGE were transferred to PVDF membranes (Millipore, Cat#PVH00010). After blocking with 5% (w/v) nonfat dry milk, the membranes were incubated with anti-FLAG antibody (Agilent, Cat# 200474-21) or anti-β-actin (Cell Signaling, Cat# 4970 S) at 4 °C overnight, which were detected with HRP-conjugated goat anti-rat immunoglobulin-G at 1:5000 dilution (Cell Signaling Technology, Cat# 7077S) or goat anti-rabbit immunoglobulin-G at 1:3000 dilution

(Cell Signaling Technology, Cat# 7074S) respectively using the chemiluminescence reagent (VWR, Cat#RPN2232). The images of the blots were taken using a Bio-Rad ChemiDoc Imaging System.

**Metal transport assay for $Zn^{2+}$ and $Cd^{2+}$.** Twenty hours post transfection, cells were washed with the washing buffer (10 mM HEPES, 142 mM NaCl, 5 mM KCl, 10 mM glucose, pH 7.3) followed by incubation with Chelex-treated culture media (DMEM plus 10% FBS). ICP-MS analysis indicated that the treatment with Chelex-100 reduced Zn content by more than 97%, whereas it had little effects on the contents of Fe and Mn (Supplementary Table 3). The Chelex-treated culture media used for transport assay already contains 0.091 μM Zn, 3.702 μM Fe, and 0.043 μM Mn. Indicated amount of metals, including the radioactive isotopes, were added into this culture media to initiate transport. To reduce the systematic errors caused by variations in transporter expression level and in cell number during transfer and wash among different samples, we added both $Zn^{2+}$ and $Cd^{2+}$ to the same cell sample by taking the advantage of $^{65}Zn$ and $^{109}Cd$ emitting gamma ray at different energy levels (1100 keV and 88 keV for Zn and Cd, respectively). Accordingly, we were able to measure the two isotopes simultaneously when two detection windows were used in the gamma counter (800–1500 keV for Zn window and 30–150 keV for Cd window). As $^{65}Zn$ also emits gamma rays within the Cd window, a calibration was performed to determine the signal solely derived from $^{109}Cd$ by subtracting the contribution of $^{65}Zn$ from the reading recorded within the Cd window. To do so, the signals of a series of $^{65}Zn$ standard samples (0–20 μM) were measured within both the Zn-window and the Cd-window. The resulting slope of the linear correlation 0.235 was used to calibrate the readings recorded within the Cd window to obtain the signals specific to Cd (Supplementary Fig. 2b). In the experiments shown in Fig. 1, 5 μM $ZnCl_2$ (0.05 μCi/well) and 5 μM $CdCl_2$ (0.05 μCi/well) were used in the substrate mixture. In the experiments shown in Figs. 3 and 8, as the variants of ZIP8 or ZIP4 and its variants exhibited low Cd transport activity, more Cd (15 μM $Cd^{2+}$, 0.15 μCi/well) was used while $Zn^{2+}$ concentration was kept the same as before. After incubation at 37 °C for 30 min, the plates were transferred on ice and an equal volume of the ice-cold washing buffer containing 1 mM EDTA was added to the cells to terminate metal uptake[11,23,36,38,40,43,46,56,61,62]. The cells were washed twice and pelleted through centrifugation at $75 \times g$ for 5 min before lysis with 0.5% Triton X-100. A Packard Cobra Auto-Gamma counter was used to measure radioactivity. The Zn (or Cd) transport activity was determined by subtracting the radioactivities of $^{65}Zn$ (or $^{109}Cd$) associated with the cells transfected with the empty vector from those associated with the cells transfected with metal transporters.

For kinetic studies, equal amount of $Zn^{2+}$ and $Cd^{2+}$ were mixed in the media at the indicated concentrations with the following radioactivity: 0.01 μCi/μM for $Cd^{2+}$ and 0.005 μCi/μM for $Zn^{2+}$. The remaining procedure was the same as described above.

**$Fe^{2+}$ transport assay.** Twenty hours post transfection, cells were washed with the washing buffer (10 mM HEPES, 142 mM NaCl, 5 mM KCl, 10 mM glucose, pH 7.3) followed by incubation with Chelex-treated DMEM plus 10% FBS. Indicated amount of $FeCl_3$, including the radioactive $^{55}Fe^{3+}$ (0.1 μCi/μM), were added into the media supplemented with 2 mM freshly prepared ascorbic acid (to reduce $Fe^{3+}$ to $Fe^{2+}$) to initiate transport[29,63]. The other procedures were the same as the Zn transport assay except that the cell lysates were mixed with the Ultima Gold cocktail (PerkinElmer, Cat# L8286) and $^{55}Fe$ associated with the cells was quantified using a liquid scintillation counter (LS 6500 Multi-Purpose Scintillation Counter).

**$Mn^{2+}$ transport assay.** Twenty hours post transfection, cells were washed with the washing buffer (10 mM HEPES, 142 mM NaCl, 5 mM KCl, 10 mM glucose, pH 7.3) followed by incubation with Chelex-treated DMEM plus 10% FBS. $MnCl_2$ was added to the media to the final concentration of 50 μM. The other procedures were the same as the Zn transport assay except that the cell lysates were digested with nitric acid and applied to Mn measurement using ICP-MS. More specifically, 75 μL of the cell lysate was mixed with 100 μL of 70% nitric acid (Fisher chemical, Cat# A509P212) in 15 mL metal-free tube (Labcon, Cat# 3134-345-001-9). The samples were heated in 60 °C water bath for 1 h to make sure the sample is fully digested. After incubation each sample was diluted to 3 mL using MilliQ water to a final solution of 2.33% nitric acid (v/v). Samples were analyzed using the Agilent 8900 Triple Quadrupole ICP-MS equipped with the Agilent SPS 4 Autosampler. Instrument calibration was accomplished by preparing standards and internal controls. The standard was IV-ICPMS-71A (Inorganic Ventures), standard concentrations were 100, 50, 25, 12.5, 6.25, 3.125 ng/mL for each element. IV-ICPMS-71D (Inorganic Ventures) was used for internal control. The relative level of Mn was expressed as the molar ratio of Mn and phosphorus[64].

**Time course experiments of metal transport assay.** To determine the optimal incubation time, time course experiments were conducted at the indicated time in the range of 0–90 min. In these experiments, the concentrations of $Zn^{2+}$, $Cd^{2+}$, and $Fe^{2+}$ were 5, 5, and 20 μM, respectively. The result is shown in Supplementary Fig. 1.

**Competition assay with $Fe^{2+}$ or $Mn^{2+}$.** For competition assay using $Fe^{2+}$ or $Mn^{2+}$, all the procedure was the same as described above, except that the incubation buffer (Chelex-treated culture media containing DMEM plus 10% FBS) contained 5 μM Zn (0.05 μCi/well) with 100 μM $FeCl_3$ plus 2 mM ascorbic acid or 100 μM $MnCl_2$.

**Immunofluorescence imaging.** HEK293T cells were grown in 24-well trays for 16 h on Poly-D-Lysine (Corning, Cat#354210) coated coverslips and transfected with plasmids harboring ZIP8 gene or the 4M variant using lipofectamine 2000. To visualize cell surface expressed ZIP8 or the 4M variant, cells were washed by Dulbecco's phosphate-buffered saline (DPBS) after 24 h transfection and then fixed for 15 min at room temperature by using 4% formaldehyde. The cells were washed by DPBS for three times and then incubated in 2% BSA in DPBS for 1 h. The cells were incubated with 4 μg/mL anti-FLAG antibody diluted with 2% BSA in DPBS at 4 °C overnight. After washing three times by DPBS, the cells were incubated with Alexa-488 goat anti-rat antibodies at 1:200 (Thermo Fisher Scientific, Cat#A11006) diluted in DPBS with 2% BSA for 1 h. After three times washing by DPBS, coverslips were mounted on the slides with fluoroshield mounting medium with DAPI (Abcam, Cat# ab104139). Images were taken with a 40 x objective using Nikon C2 confocal microscope. To detect the overall expression of ZIP8 and the 4M variant, after fixation by 4% formaldehyde, cells were permeabilized and blocked for 1 h with DPBS containing 2% BSA and 0.1% Triton X-100 and then were incubated with 4 μg/mL anti-FLAG antibody diluted with 2% BSA in PBST with 0.1% Tween-20 (ThermoFisher Scientific, Cat# 85113) at 4 °C overnight. The other procedures were the same as those for the non-permeabilized cells.

**Bioinformatics and structural modeling.** Multiple sequence alignment was conducted using Clustal Omega (https://www.ebi.ac.uk/Tools/msa/clustalo/). A threshold of 50% sequence identity was used as the cutoff to select orthologs of ZIP8, ZIP14, and ZIP4. Structure models were generated by SWISS MODEL (https://swissmodel.expasy.org/interactive). The structural model of human ZIP8 in the IFC was generated using the BbZIP structure (PDB: 5TSB) as the template. The models of ZIP8 in the OFC were created using either the OFC model of BbZIP[65] or the AlphaFold predicted human ZIP13 (https://alphafold.ebi.ac.uk/entry/Q96H72) as the template. Other ZIPs were modeled using the AlphaFold predicted human ZIP13 as the template. Evolutionary covariance analysis was conducted by using EVcouplings (https://v2.evcouplings.org/). To study the amino acid composition at the positions where Q180 and E343 are located in ZIP8, the protein sequences of the PF02535 family in the Pfam database (https://pfam.xfam.org/family/Zip) were retrieved, classified, and analyzed using the approach described previously[14]. The aligned sequences in each subfamily were analyzed in Jalview.

**MD simulation and free energy calculation.** MD simulations were performed using AMBER20[66]. The system was prepared using CHARMM-GUI[67], in which the box dimension was $101 \times 101 \times 126$ Å containing one ZIP8 protein, 262 1,2-dilauroyl-sn-glycero-3-phosphocholine (DLPC) molecules, and 25973 OPC water molecules. Minimization was done in five stages with a gradient of restriction from protein backbone to side chain, each step yield 10,000 steps of steepest descendent and 10,000 steps of conjugate gradient methods; then 36 ns of NVT heating was performed with the temperature increasing gradually from 0 to 300 K. Then another 3 μs of simulation was performed to equilibrate the system in the NPT ensemble. Finally, a seven-window thermodynamic integration was conducted on the equilibrated system, each window yielded 300 ns. For every 50 ps the snapshot was saved to the trajectory file, yielding 420, 000 snapshots for the Gaussian Quadrature analysis. 10 Å cutoff was used for the non-bonded interaction. PME method and PBC were used for the simulations, and the Langevin algorithm with a 2.0 $ps^{-1}$ friction coefficient was used for maintaining the temperature[68]. Berendsen barostat was used for pressure control with a relaxation time of 1.0 ps[69]. The time step was 1.0 fs with SHAKE used to constrain the bonds containing hydrogen atoms[70].

To compare free energy changes upon metal ions binding to the selectivity filter involving the side chains of H180, H343, D318, and E184 in the 4M variant, a modified 12-6-4 Lennard-Jones (m-12-6-4-LJ) non-bonded model was introduced to consider the induced dipole effect between the metal ion and coordinating atoms during simulation[71]. Since the induced dipole effect is highly dependent on the polarizability of coordinating atoms, all parametrized polarizabilities used in this research was provided by the table in Supplementary Fig. 5a. Because the m-12-6-4-LJ model mainly yields strongly localized interactions, only the residues at the metal binding sites were applied with modified polarizabilities and m-12-6-4-LJ potentials.

**Statistics and reproducibility.** Statistical analysis was conducted using the two-sided Student's $t$ test. Sample information for the biochemical data shown in figures in the main text or in SI is detailed in the figure legends. When the results of a representative experiment are shown in a figure, the other two independent experiments showed similar results. Measurements of replicates were taken from distinct samples.

**Reporting summary**. Further information on research design is available in the Nature Portfolio Reporting Summary linked to this article.

## Data availability

The structural data cited in this study are available under accession codes 5TSB in PDB. The AlphaFold predicted human ZIP13 structure is retrieved from https://alphafold.ebi.ac.uk/entry/Q96H72. The source data underlying Figs. 1, 3–5, and 8 are provided in Supplementary Data 1 (source data_updated.xlsx). The uncropped Western blots presented in Fig. 4a are shown in Supplementary Fig. 7.

## Code availability

The input and final output files can be accessed via the PMEMD.cuda program in AMBER20 (a license is required) at the following link with free access (https://github.com/lizhen62017/hZIP8). The "README.md" file in the GitHub repository contains explanations of the output files and instructions for reproducing the results.

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

## Acknowledgements
We thank Dr. Jon Kaguni at Biochemistry & Molecular Biology in Michigan State University for instructing us to use the liquid scintillation counter. We thank Dr. Thomas O'Halloran at MMG in Michigan State University for the guidance of using ICP-MS. We thank Dr. Erik Shapiro at Department of Radiology in Michigan State University for the insightful discussion of manganese transport assay. This work is supported by NIH GM129004 and GM140931 (to J.H.), the MSU Plant Resilient Institute seed grant (to H.T. and J.H.), NIH GM130641 (to K. Merz). K. MacRenaris is supported by NIH GM135018, GM115848, GM038784 (to Thomas V. O'Halloran).

## Author contributions
J.H. conceived project and designed the study. Y.J., D.S., Z.L., G.S., T.W., and K. MacRenaris conducted experiments and/or computational studies. Y.J., K. Merz, H.T., and J.H. analyzed the data and wrote the manuscript.

## Competing interests
The authors declare no competing interests.
