## [Peer Review File · Communications Biology]

Reviewers' comments:

Reviewer #1 (Remarks to the Author):

This contribution addresses the molecular determinants of metal selectivity by ZIP transporters. In particular, the authors developed a quadruple mutant of ZIP8 with increased selectivity for zinc(II). The properties of the mutant have been characterized through a combination of experimental methods, in addition to molecular dynamics simulations. The overall approach is sound and the results are convincing.

My only comment regards the focus on ZIP8/ZIP4/ZIP14. The authors state that "ZIP8 is a member of the LIV-1 subfamily including nine human ZIPs", where ZIP8 and ZIP14 are involved in uptake of cadmium(II) whereas elsewhere it is mentioned that ZIP4 is a member of the family that prefers zinc(II). What about the other members of the family - why were they not taken into account for the determination of DCRs? This is an important point for the definition of the target residues, thus it must be explained more in detail.

Reviewer #2 (Remarks to the Author):

This is an interesting contribution with novel data from two groups that are internationally well recognized for their contributions in structural and computational biology. The manuscript deals with the issue of metal selectivity of a subgroup of zinc transporters that also transport cadmium, iron, and manganese. The transporter addressed here is Zip8 (SLC38A8). Employing structure modelling, evolutionary covariance, mutagenesis, and transport assays, the group identified two residues that are important for the selectivity. The work has implications for basic science and applied science (bioengineering).

The following issues need to be addressed:

A statement is necessary that the only available X-ray structure is from a bacterial protein and that this structure is serving for discussions of structure/function relationship of this family of 14 human transporters. The elevator mechanism of the bacterial protein was discussed last year in papers from the Hu group in Nature Comm (not cited, but BioRxiv reference given instead) and from a group in Denmark in Science Advances. It is important to explain the history of developments in the field to the reader, namely that the binuclear metal transport site described is not established for the human proteins by structural studies. This point becomes even more important when the participation of two cysteines in this site is described as having been established for Zip8 and Zip14.

It is not clear why this article focuses only on Zip8 and does not discuss the implications for the closely related transporter Zip14. Discussion of Zip14 should be included.

The abstract is unclear regarding the relation to Zip4 and the major finding of a new filter without giving the residues involved. The abstract needs work. Also, in the introduction, the residue pair is mentioned, but again without naming the amino acids identified.

Why do the authors use the term selective filter instead of the more commonly used term selectivity filter?

Some discussion is necessary whether this transporter is a channel and whether metal binding is the only factor that drives the conformational change. What energizes the elevator?

On p. 2, the authors write that the entire family controls zinc, iron, and manganese. It is not correct, many of the transporters have selectivity for zinc. The authors should provide more information on the literature that has linked the two transporters, Zip8 and Zip14, to manganese and iron metabolism under some conditions, and they should cite work of those investigators that have made the original and key observations. Is it just a case of promiscuity, or are these transporters participating in specific aspects of iron and manganese metabolism as other transporter for these metal ions exist?

The figure about the sequence alignments in the supplementary material should be in the main part. It is important information for reference and for being able to follow the text.

The authors give a distance range of 7.5 to 8.9 Å for the pair Q180 – E343. To which atoms does the

distance refer? Without this knowledge one should not call it a physical interaction. Epistasis is a concept in genetics. If employed here in structural biology, it should be explained. Further information on the transport assays and their interpretation is necessary. The authors talk about the substrate as the naked metal ion. This is not correct. The ions are either hydrated or have other ligands such as chloride and the ligand exchange of such complexes can be part of the selectivity mechanism as suggested for other metal ion transporters. It is not quite clear why Fe(III) and ascorbate were used instead of an iron(II) salt such as iron ammonium sulphate, which is not readily oxidized. Is the substrate then a Fe(II) – dehydroascorbate complex? The authors say that they stopped the reaction with EDTA. This is problematic as the Zn/EDTA complex has been reported to be a better substrate than Zn itself in uptake assays. The assay is described as having 10% FBS present. FBS contains zinc, iron, and manganese. The concentrations need to be known for the evaluation of the kinetic data as the transport assay does not start with zero concentrations of the metal ions in the presence of FBS. The authors use a Hill equation. They need to give the Hill coefficient and discuss possible reasons for cooperativity. Vmax should be reported as a relative number only if the absolute rate is known and stated. What is it? Were zinc and cadmium used as their respective chloride salts? Why was the DMEM treated with Chelex-100? Are the ratios given Vmax ratios?

I also do not understand Fig. S1. The data points are not fitted and do not extrapolate to zero. It is relative radioactivity. How does radioactivity relate to concentrations? Knowing this relationship, one would have comparative rates for the different metal ions.

Values for centrifugation in rpm are relative and need to be expressed as $\times g$.

On p. 3, it says "the same cells." Which cells? HEK293T cells are mentioned only later. Is it the only cell line used?

Reviewer #3 (Remarks to the Author):

In this manuscript, Jiang et al reported that the substrate preference of ZIP8 with multiple metal substrates could be modified by replacing its differently conserved amino acid residues with the counterparts in zinc-preferring ZIP4. They showed that the quadruple variant of ZIP8 exhibited reduced transport activities towards Cd, and also Fe and Mn, whereas increased activity toward Zn. Moreover, they showed that a conditional selective filter functions only when the transporter adopts the outward-facing conformation, using structural modeling and MD simulations. Experiments are well done and reasonably performed, but there are several pointed to be improved.

Comments.

The primary and critical question is that authors did not show whether the reverse quadruple substitution conducted on other ZIPs, such as ZIP4, showed multiple preference to Cd, Fe and Mn, in addition to Zn (alternatively, introduction of Q180 and E343 pair into ZIP4). Although authors clearly demonstrated that the quadruple substitution on ZIP8 is necessary condition for the zinc-preferring of ZIP8 to Cd, Fe and Mn, the reverse confirmation is critical for engineering of ZIPs. Although no direct evidences had not been presented thus far, unique features of Q180 and E343 of ZIP8 and ZIP14 has been thought to be associated with their unique metal specificity. Therefore, the reverse experiments on zinc-preferring ZIPs should be added to clearly reveal the importance of the finding shown here.

Other comments.

1). Did ZIP8 mutants generated in this study showed the same cell surface expression levels? Their expression was shown only in the simple Western blot. Some mutants showed significant reduction of the transport ability. Therefore, more detailed examination is required for clarification (e.g. such as biotinylation).

2). Considering that E343H ZIP8 mutant caused loss of the metal transport activity, does the Q180 have block the access of metals on the entrance? As shown in Fig. 5C, many members of ZIP_{II}, ZIP9 and GufA/ZupT subfamily have other residues, not His residue, at the position corresponding to Q180. How do ZIPs generally recognize and discriminate the substrate metal using this position?

3). In Figure S5, 4M ZIP8 mutant looks to still has Cd transport ability compared with ZIP4. Are there any other residues involved in Cd recognition in ZIP8?

Response to Review

Note: Reviewers' comments in black, and our responses in blue.

Reviewers' comments:

Reviewer #1 (Remarks to the Author):

This contribution addresses the molecular determinants of metal selectivity by ZIP transporters. In particular, the authors developed a quadruple mutant of ZIP8 with increased selectivity for zinc(II). The properties of the mutant have been characterized through a combination of experimental methods, in addition to molecular dynamics simulations. The overall approach is sound and the results are convincing.

R: We thank Reviewer#1's positive comments.

My only comment regards the focus on ZIP8/ZIP4/ZIP14. The authors state that "ZIP8 is a member of the LIV-1 subfamily including nine human ZIPs", where ZIP8 and ZIP14 are involved in uptake of cadmium(II) whereas elsewhere it is mentioned that ZIP4 is a member of the family that prefers zinc(II). What about the other members of the family - why were they not taken into account for the determination of DCRs? This is an important point for the definition of the target residues, thus it must explained more in detail.

R: Thanks for asking this important question. We identified DCRs in two rounds of sequence comparison. In the first round, we compared ZIP8 and ZIP14 with all other LIV-1 members (**Figure 1A**). Only LIV-1 family members were included is because the ZIPs in other subfamilies (ZIP1-3 in ZIPII, ZIP9 in ZIPI, and ZIP11 in GufA) are evolutionarily distant from the LIV-1 family members. In general, comparing the proteins with higher sequence identity will have a better chance of identifying the key residues responsible for a feature than comparing those with lower sequence identity, if the former group has already diverged with respect to the feature of interest. For this reason, the ZIPs in other subfamilies were not included to identify DCRs. However, we don't think these two residues are the only ones that determine whether a LIV-1 is a zinc-preferring or multi-metal transporter. Consistently, the double variant (Q180H/E343H) can still transport Cd, although with a lower transport rate. In order to identify the missing DCRs, we focused on the comparison of ZIP8/14 with ZIP4 in the second round because ZIP4 share the highest sequence identity with ZIP8/14 than other LIV-1 members (**Table S1**) and also ZIP4 is a well-characterized zinc transporter.

The follow paragraph is added in the revised manuscript.

"We chose ZIP4 to compare ZIP8 and ZIP14 is because ZIP4 shares the highest sequence identity with ZIP8/14 than other LIV-1 members (Table S1) and also it is a well-characterized zinc transporter."

Reviewer #2 (Remarks to the Author):

This is an interesting contribution with novel data from two groups that are internationally well

recognized for their contributions in structural and computational biology. The manuscript deals with the issue of metal selectivity of a subgroup of zinc transporters that also transport cadmium, iron, and manganese. The transporter addressed here is Zip8 (SLC38A8). Employing structure modelling, evolutionary covariance, mutagenesis, and transport assays, the group identified two residues that are important for the selectivity. The work has implications for basic science and applied science (bioengineering).

The following issues need to be addressed:

1. A statement is necessary that the only available X-ray structure is from a bacterial protein and that this structure is serving for discussions of structure/function relationship of this family of 14 human transporters. The elevator mechanism of the bacterial protein was discussed last year in papers from the Hu group in Nature Comm (not cited, but BioRxiv reference given instead) and from a group in Denmark in Science Advances. It is important to explain the history of developments in the field to the reader, namely that the binuclear metal transport site described is not established for the human proteins by structural studies. This point becomes even more important when the participation of two cysteines in this site is described as having been established for Zip8 and Zip14.

R: Thanks for the suggestion. More background information has been added in the second paragraph of the section of Introduction.

2. It is not clear why this article focuses only on Zip8 and does not discuss the implications for the closely related transporter Zip14. Discussion of Zip14 should be included.

R: Due to our limited capability, we have not expanded this study to ZIP14, which is another very interesting multi-metal transporter in the same LIV-1 subfamily. Given the high sequence identity between ZIP8 and ZIP14 (48%), we postulate that the same mutations in ZIP14 would generate the same or similar effects on the substrate specificity, although it needs to be examined experimentally. We discussed this possibility in Discussion as follows.

“Given the high sequence identity between ZIP8 and ZIP14 (48%, Table S1), the same mutations in ZIP14 may generate similar effects on changing substrate specificity. After all, the importance of the proposed selectivity filter has been confirmed in ZIP4 (Figure 8), which shares a lower sequence identity (31%, Table S1). Whether or not the identified residue pair plays a similar role in ZIPs with longer evolutionary distance needs to be tested in future study.”

3. The abstract is unclear regarding the relation to Zip4 and the major finding of a new filter without giving the residues involved. The abstract needs work. Also, in the introduction, the residue pair is mentioned, but again without naming the amino acids identified.

R: We have updated the abstract. Due to the limit on word number (150 words), we couldn't add more details in the abstract but included additional background information in Introduction, including the information of the quadruple variant and the residue pair.

5. Why do the authors use the term selective filter instead of the more commonly used term selectivity filter?

R: Both terms are used in literature. As suggested by the reviewer, we replaced it to a more commonly used “selectivity filter”.

6. Some discussion is necessary whether this transporter is a channel and whether metal binding is the only factor that drives the conformational change. What energizes the elevator?

R: As suggested by the reviewer, we added the following sentence in Discussion.

“..., which is a likely carrier according to the saturable kinetic profile (Figures 4 & 5), ...”

The energy source to elicit conformational changes for ZIPs are not identical based on literature. For instance, ZIP8/14 is reported to co-transport metal with bicarbonate, whereas ZIP4 co-transport zinc with protons. As the transport mechanism is not the focus of this work, we did not include this in Discussion.

7. On p. 2, the authors write that the entire family controls zinc, iron, and manganese. It is not correct, many of the transporters have selectivity for zinc. The authors should provide more information on the literature that has linked the two transporters, Zip8 and Zip14, to manganese and iron metabolism under some conditions, and they should cite work of those investigators that have made the original and key observations. Is it just a case of promiscuity, or are these transporters participating in specific aspects of iron and manganese metabolism as other transporter for these metal ions exist?

R: We did not mean that every ZIP transports Zn, Fe, and Mn and we have rephrased the mentioned sentences to clarify this point. The relevant sentence has been modified as follows.

“..., while most of the fourteen human ZIPs are reported to transport Zn^{2+} and play roles in Zn homeostasis and Zn signaling, ZIP8 and its close homolog ZIP14 transport not only Zn^{2+} but also ferrous ions (Fe^{2+}), manganese ions (Mn^{2+}), and cadmium ions (Cd^{2+}), and as such are critically involved in Fe and Mn homeostasis and are responsible for cellular Cd uptake and toxicity.”

We apologize for missing key references. More relevant references about ZIP8 and ZIP14 transporting Fe and Mn have been cited in Introduction.

The physiological functions of ZIP8 and ZIP14 as manganese and iron transporters have been thoroughly reviewed in ref 24 and ref 28, respectively.

8. The figure about the sequence alignments in the supplementary material should be in the main part. It is important information for reference and for being able to follow the text.

R: Thanks for this suggestion. We have moved the sequence alignments (the previous Figure S4) to the main text as Figure 2.

9. The authors give a distance range of 7.5 to 8.9 Å for the pair Q180 – E343. To which atoms does the distance refer? Without this knowledge one should not call it a physical interaction. Epistasis is a concept in genetics. If employed here in structural biology, it should be explained.

R: The mentioned distance refers to the distance between two C α atoms of Q180 and E343, which is indicated in the legend of Figure 6A. According to Figures 6D and 7A, Q180 and E343 are able to form a direct physical contact.

To better explain epistasis in the context of structural biology, the following sentence, including a review on epistasis, is added in Discussion.

“Physical contact of the involved residue pair is one of many reasons that explain the epistasis of two compensatory mutations (ref 59).”

In ref 59, the section entitled “Specific Epistasis Due to the Three-Dimensional Structure of Molecules” makes a thorough discussion on this topic.

10. Further information on the transport assays and their interpretation is necessary. The authors talk about the substrate as the naked metal ion. This is not correct. The ions are either hydrated or have other ligands such as chloride and the ligand exchange of such complexes can be part of the selectivity mechanism as suggested for other metal ion transporters.

R: We agree with the reviewer’s comment. In the revised manuscript, we add one sentence in Introduction to clarify this point.

“Note that although the exact metal species that is transported by ZIPs has not been fully elucidated, M^{2+} is used to indicate that the ZIPs transport divalent metal substrates.”

Another sentence is added in Discussion.

“Uniquely for metals, their interactions with a variety of ligands in solution (water, counterions, and other small molecule ligands) should also be considered, as the ligand exchange rates of metal-ligand complexes can affect metal selectivity.”

11. It is not quite clear why Fe(III) and ascorbate were used instead of an iron(II) salt such as iron ammonium sulphate, which is not readily oxidized. Is the substrate then a Fe(II) – dehydroascorbate complex?

R: We followed the conditions used to study other Fe(II) transporters, including IRT1, a plant Fe(II) transporter, and DMT1, a mammalian Fe(II) transporter (refs 29 and 63). When purchasing radioactive ^{55}Fe , we could only find $^{55}\text{Fe(III)Cl}_3$. We agree that Fe(II) is likely in multiple forms, including the suggested Fe(II)-dehydroascorbate complex. Our results and those of others showed that $^{55}\text{Fe(II)}$ was able to be transported into the cells under this condition.

12. The authors say that they stopped the reaction with EDTA. This is problematic as the Zn/EDTA complex has been reported to be a better substrate than Zn itself in uptake assays.

R: Adding EDTA to stop transport reaction is a widely and frequently used approach to study the ZIP-mediated transport in the cell-based assays to remove zinc ions bound at the cell surface (refs 11, 23, 36, 38, 40, 43, 46, 56, 61, 62). In particular, an early study of human ZIP1 (ref 61) investigated whether or not PC-3 cells (a human prostate cancer cell line) take up zinc chelated with EDTA and their results clearly demonstrated that the cells did not uptake ^{65}Zn (20 μM) in the presence of 60 μM of EDTA in 15 minutes (Figure 7A in their paper).

13. The assay is described as having 10% FBS present. FBS contains zinc, iron, and manganese. The concentrations need to be known for the evaluation of the kinetic data as the transport assay does not start with zero concentrations of the metal ions in the presence of FBS.

R: We analyzed the contents of Zn, Fe, and Mn before and after the treatment of the culture media (DMEM+10% FBS) with Chelex-100 resin. The results (Table S3) showed that Chelex-100 resin removed more than 97% of Zn in the culture media but had much smaller effects on Mn or Fe. As the residual Zn level (<0.1 μM) is much lower than the measured K_M (1-3 μM , Figure 4), we did

not make adjustment in data processing or curve fitting. For the same reason, we did not change the figures for the transport assays of Fe or Mn. Instead, we report the ICP-MS results (Table S3) and clarified that there are residual metals in the Chelex-treated culture media under “*Metal transport assay*” in the section of Methods.

14. The authors use a Hill equation. They need to give the Hill coefficient and discuss possible reasons for cooperativity. V_{max} should be reported as a relative number only if the absolute rate is known and stated. What is it? Were zinc and cadmium used as their respective chloride salts? Why was the DMEM treated with Chelex-100? Are the ratios given V_{max} ratios?

R:

The Hill coefficients have been added in the updated figures (Figures 4C & 5B). The following sentences are added in the section of transport kinetic study.

“The positive cooperativity as indicated by the Hill coefficients ($n=1-2$) may result from the BMC (M1 and M2) in the transport site and/or from the interactions between the two monomers of the ZIP dimer. Dimerization seems to be a common feature among the ZIP family members.”

The unit of V_{max} is radioactivity/minute. For comparison and statistical analysis of data from different independent experiments, V_{max} was normalized to the V_{max} of the wild-type protein that was obtained in the same batch of experiment.

Zinc and cadmium were used as chloride salts. This information has been added in the revised manuscript.

Chelex-100 treatment is to reduce transition metals from the culture media (DMEM+FBS) so that the results would be less affected due to the varied metal contents in different batches of FBS.

The Zn/Cd selectivity (Figures 1, 3, 8) are not the ratios of V_{max} , they are the ratios of the radioactivities of ^{65}Zn and ^{109}Cd measured simultaneously in the same sample. This ratio reflects the relative transport rate of Zn and Cd.

15. I also do not understand Fig. S1. The data points are not fitted and do not extrapolate to zero. It is relative radioactivity. How does radioactivity relate to concentrations? Knowing this relationship, one would have comparative rates for the different metal ions.

R: Fig. S1 shows the linear relationship in the time course experiments, indicating that the chosen reaction time in our experiments (30 min) is appropriate and the data collected under this condition can be used to calculate the initial transport rate for kinetic analysis. We want to clarify that the data shown in Fig. S1 are the absolute readings of radioactivity (raw and unprocessed data).

16. Values for centrifugation in rpm are relative and need to be expressed as $\times g$.

R: rpm has been converted to $\times g$.

17. On p. 3, it says “the same cells.” Which cells? HEK293T cells are mentioned only later. Is it the only cell line used?

R: We removed the confusing word “same”. The cells are HEK293T.

Reviewer #3 (Remarks to the Author):

In this manuscript, Jiang et al reported that the substrate preference of ZIP8 with multiple metal substrates could be modified by replacing its differently conserved amino acid residues with the counterparts in zinc-preferring ZIP4. They showed that the quadruple variant of ZIP8 exhibited reduced transport activities towards Cd, and also Fe and Mn, whereas increased activity toward Zn. Moreover, they showed that a conditional selective filter functions only when the transporter adopts the outward-facing conformation, using structural modeling and MD simulations. Experiments are well done and reasonably performed, but there are several pointed to be improved.

R: We appreciate the reviewer's positive comments.

Comments.

The primary and critical question is that authors did not show whether the reverse quadruple substitution conducted on other ZIPs, such as ZIP4, showed multiple preference to Cd, Fe and Mn, in addition to Zn (alternatively, introduction of Q180 and E343 pair into ZIP4). Although authors clearly demonstrated that the quadruple substitution on ZIP8 is necessary condition for the zinc-preferring of ZIP8 to Cd, Fe and Mn, the reverse confirmation is critical for engineering of ZIPs.

Although no direct evidences had not been presented thus far, unique features of Q180 and E343 of ZIP8 and ZIP14 has been thought to be associated with their unique metal specificity. Therefore, the reverse experiments on zinc-preferring ZIPs should be added to clearly reveal the importance of the finding shown here.

R: Thanks for this great suggestion. We conducted the suggested experiments on human ZIP4. The results are shown in Figure 8 and the following paragraph has been added in the revised manuscript.

*“To test the importance of the selectivity filter in another ZIP, we chose human ZIP4 to perform reverse substitution on the residues that are topologically equivalent to those that were mutated in the 4M variant of ZIP8. As shown in **Figure 8A**, for Zn²⁺ and Cd²⁺ transport, the results indicated that: (1) The H379Q mutation (the reverse substitution of Q180H in ZIP8) showed little effect on Zn²⁺ or Cd²⁺ transport activity; (2) The H536E mutation (the reverse substitution of E343H in ZIP8) completely abolished the Zn²⁺ transport activity but retained a marginal activity toward Cd²⁺; (3) Combining the H379Q and H536E mutations (the reverse substitutions of the 2M variant) partially restored Zn²⁺ and Cd²⁺ transport activities and the Zn/Cd selectivity was significantly reduced by more than 60%, mirroring the effect of the 2M variant of ZIP8 that exhibits an increased Zn/Cd selectivity; and (4) Incorporation of additional two mutations (G503C and H550N, the additional two reverse substitutions of the 4M variant) with the H379Q/H536E variant abrogated Zn²⁺ and Cd²⁺ transport activities. Importantly, the partial restoration of Zn²⁺ and Cd²⁺ activities of the H536E variant by the H379Q mutation suggests an epistatic interaction, consistent with the proposed structural model where residues at these two positions are in close proximity when the transporter is in the OFC (**Figure 6**). The significantly reduced Zn/Cd selectivity of the H379Q/H536E variant reinforces the notion that these two residues at the selectivity filter play a role in determining substrate specificity. As the H379Q mutation had little effect on Zn²⁺ or Cd²⁺*

transport activity, it seems that the H536E mutation is responsible for the reduced Zn/Cd selectivity. Indeed, the single H536E mutation did not completely abrogate Cd²⁺ transport activity as it did for Zn²⁺ transport. While we did not detect Fe²⁺ transport activity for any of the tested constructs of ZIP4, a Mn²⁺ transport activity was detected for the wild-type ZIP4 and the H379Q variant (Figures 8B & C). The Mn²⁺ transport activity was diminished by the H536E mutation but could not be restored by the H379Q mutation. Interestingly, the quadruple variant (H379Q/H536E/G503C/H550N) exhibited a partially restored Mn²⁺ activity. Overall, the mutagenesis study on ZIP4 confirmed the importance of the selectivity filter in determining substrate preference, whereas the other two mutations along the transport pathway appeared to function differently from their counterparts in ZIP8.

Other comments.

1). Did ZIP8 mutants generated in this study showed the same cell surface expression levels? Their expression was shown only in the simple Western blot. Some mutants showed significant reduction of the transport ability. Therefore, more detailed examination is required for clarification (e.g. such as biotinylation).

R: We conducted immunofluorescence experiments to detect cell surface expression of the generated ZIP8 variants. As shown in Figure S3C, all the variants are expressed at the cell surface, indicating that they have no severe defects in trafficking and that the changed radioactivity associated the cells in the transport assay is caused by the altered transport activity.

2). Considering that E343H ZIP8 mutant caused loss of the metal transport activity, does the Q180 have block the access of metals on the entrance? As shown in Fig. 5C, many members of ZIPII, ZIP9 and GufA/ZupT subfamily have other residues, not His residue, at the position corresponding to Q180. How do ZIPs generally recognize and discriminate the substrate metal using this position?

R: We don't think Q180 blocks the entrance based on the structural model (Figure 6D). Q180 is neither big enough nor electrostatically repulsive to prevent metals from entering the pore.

As shown in Figure 6D (the previous Figure 5D), there are three or four metal chelating residues that are conserved in each subfamily at the pore entrance. Therefore, it is likely that, even the position equivalent to Q180 in ZIP8 is not highly conserved (for instance, it is occupied by a hydrophobic residue in ZIP9), other residues may still form the proposed selectivity filter. To clarify this point, the following sentence is added in the revised manuscript.

"..., although the amino acid composition of this metal binding site may vary in different subfamilies (Figure 6D)."

In addition, we do not mean that Q180 or the proposed selectivity filter is the only factor that determines substrate specificity of the ZIPs. Additional mechanisms must exist. The following sentence is added to clarify this point.

"Also unlike the selectivity filters of canonical ion channels, the proposed selectivity filter of ZIP8 must not be the only mechanism that determines substrate specificity, since residues in the transport site and along the transport pathway (see below) are the obvious candidates involved in discriminating metal substrates."

3). In Figure S5, 4M ZIP8 mutant looks to still has Cd transport ability compared with ZIP4. Are there any other residues involved in Cd recognition in ZIP8?

R: The 4M variant is still a more promiscuous transporter than ZIP4. There must be additional mechanism(s) to allow the 4M variant to transport Cd²⁺, but we have not identified them at this point. To completely eliminate the residual Cd transport activity, additional rounds of mutagenesis and screen are needed. For example, we did not test whether the non-metal chelating DCRs identified in sequence alignment (Figure 2, previous Figure S4) play any role. The following sentence is added in the revised manuscript to clarify this point.

“However, the residual Cd²⁺ transport activity of the 4M variant suggests that there are additional unidentified mechanism(s) to allow the 4M variant to transport Cd²⁺.”